# Neprilysin 4 controls acrosome structure and male fertility in *Drosophila melanogaster*
Annika Buhr [1], Maike Spielmeyer [1], Eva Cordes[1], Leonhard Breitsprecher[2], Rico Franzkoch[2], Isabel Kipp[2], Olympia Ekaterini Psathaki[2], Stefan Walter[2], Achim Paululat[1,2] & Heiko Meyer [1,2] ✉

The acrosome is a highly conserved organelle located at the anterior tip of the mature sperm. It is essential to successful fertilization across species. Herein, we provide evidence that the peptidase Neprilysin 4 localizes to the acrosome during *Drosophila* sperm maturation and that activity of this enzyme is essential to proper acrosome structure, sperm tip partitioning and male fertility. In contrast to controls, sperm from animals with impaired Nep4 function exhibited a severely disturbed tip structure that affected intracellular and sperm surface components. These effects were secondary to structural defects in the acrosome morphology. Corresponding sperm were quickly discarded by the females, exhibited a reduced ability to fertilize eggs, and did not initiate embryonic development, even if sperm entry had occurred. This study demonstrates an essential role for a neprilysin in the structural integrity of the acrosome and the sperm tip, and thus introduces a new and crucial function of neprilysins in ensuring male fertility.

During maturation, sperm cells undergo several stages of development to eventually become motile sperm capable of fertilizing eggs. All sperm evolve from germ line stem cells and develop during spermatogenesis within the testis. In *Drosophila melanogaster*, the first half of spermatogenesis is characterized by the differentiation of stem cells into individual bundles of 64 round spermatids. Throughout the second half, termed spermiogenesis, the spermatids undergo drastic morphological changes and several intermediate stages to finally form individual motile sperm, which are stored in the seminal vesicle until ejaculation[1–3]. Upon copulation, the sperm cells are exposed to different environments during the transfer from the male to the female reproductive tract. Sperm surface composition is further modified during this period. First, seminal fluid proteins are released from male accessory glands and associate with the sperm during ejaculation. Transferred sperm additionally interact with female-derived factors in the uterus. Some of these factors are directly associated with the sperm, whereas earlier factors are processed only in the female environment. Prior to their use in fertilization, the sperm are retained in two female storage organs, the seminal receptacle and the spermatheca, in which further factors contribute to sperm composition. Certain early factors, predominantly seminal fluid proteins, disappear during storage, while the proportion of female-derived proteins associated with the sperm increases to support sperm viability[4]. These dynamic interactions of the sperm with surrounding factors are widespread across the animal kingdom and crucial for successful fertilization. The capability of sperm to fertilize eggs is thus achieved by continuous sperm modification both before and after ejaculation[5]. However, knowledge of the relevant factors and their individual molecular functions is still incomplete.

In both mammals and *Drosophila*, a sperm-specific organelle that is essential to successful fertilization is the acrosome. It emerges from the Golgi-derived acroblast during spermiogenesis and, accordingly, proper function of the Golgi and its resident proteins is crucial to the formation of mature fertile sperm[6–9].

In this study, we identified the *Drosophila* endopeptidase Neprilysin 4 (Nep4) as a novel acrosomal protein essential to fertilization. Previous investigations already showed that germ line-specific *nep4* knockdown or genetic deletions encompassing the majority of the extracellular Nep4 domain result in male sterility[10,11]. However, the molecular mechanism underlying these effects remained elusive, and detailed morphological or ultrastructural analyses of *nep4* mutant sperm have not been performed yet. Nevertheless, previous studies confirmed *nep4* expression[12] as well as protein presence[13] in different parts of the *Drosophila* testis. Moreover, neprilysins in general were identified as crucial male-specific factors that regulate egg production and sperm utilization in their respective female mating partners[12]. Significantly, in line with the results for *Drosophila nep4*, *Nep2*-deficient mice were described as viable and female fertile, yet males exhibited reduced fertility[14]. Since the testes and sperm of corresponding animals

[1]Department of Zoology and Developmental Biology, Osnabrück University, Osnabrück, Germany. [2]Center of Cellular Nanoanalytics (CellNanOs), integrated Bioimaging Facility (iBiOs), University of Osnabrück, Osnabrück, Germany. ✉e-mail: meyer@biologie.uni-osnabrueck.de

appeared normal, the effect was attributed to a potential perturbation in the sperm maturation process. However, the underlying molecular mechanism again remained unknown.

In this study, we used CRISPR/Cas9-assisted homologous recombination to generate a *Drosophila* line expressing an endogenously tagged Nep4::mNeonGreen fusion protein. The resulting animals were used to gain deeper insights into the subcellular localization of Nep4 in the testis and sperm. Moreover, we investigated the role of the protein in the sperm maturation process in detail. We found that the correct morphology of the acrosome and the sperm tip depended on the expression of functional Nep4, while the presence of an inactive Nep4 variant or germline-specific *nep4* knockdown led to significant malformations. To our knowledge, such a function in supporting the morphology of the acrosome and the sperm tip structure has not yet been described for a neprilysin and could therefore represent a decisive new aspect of male fertility across species.

## Results

### Homozygous Nep4::mNG flies are male sterile

Previous studies showed that germ line-specific *nep4* knockdown or deletion of the extracellular protein domain results in male sterility[10,11]. However, the molecular basis of this phenotype as well as the subcellular localization of Nep4 in testes or sperm remained elusive. To investigate this issue, we used CRISPR/Cas9-assisted homology directed repair to generate a fly line that expresses endogenously mNeonGreen (mNG)-tagged Nep4. We confirmed in-frame insertion of the mNG sequence at the endogenous locus by genomic DNA sequencing. Expression and stability of the tagged Nep4 construct were validated by western blot analysis (Supplementary Fig. 1). Interestingly, the resulting Nep4::mNG flies were homozygous viable but male sterile, as shown by fertility analyses. While mating control males (*w*[1118]) to wildtype virgin females resulted in an average of 36 offspring after 18 days, corresponding experiments using homozygous Nep4::mNG males (Nep4::mNG[homo]) resulted in a drastically reduced average number of offspring (0.4 offspring, Fig. 1). In contrast, the number of offspring using heterozygous Nep4::mNG males (Nep4::mNG[hetero]) was comparable to the control (31 offspring). These data indicate that the C-terminal mNG fusion results in a Nep4 protein that at least partially loses its functionality, eventually causing male sterility under homozygous conditions. Of note, neither heterozygous nor homozygous Nep4::mNG females exhibited any impairments in fertility (Supplementary Fig. 2), which confirmed that only males are affected. To further validate the specificity of the phenotype, we also analyzed *nep4*[RNAi] knockdown males for impaired fertility (using testis-specific *bam*-Gal4 as a driver). The numbers of offspring for control males (*w*[1118] x *bam*-Gal4 and *w*[1118] x UAS-*nep4*[RNAi]) were 62.6 and 67.8 on average after 18 days, respectively. However, the fertility of the *nep4*[RNAi] knockdown males (*bam* > *nep4*[RNAi]) was significantly reduced, as indicated by only 14.4 offspring. These results indicate that sterility of Nep4::mNG[homo] males is caused by reduced function of the Nep4 fusion protein in testes, and not by CRISPR/Cas9-mediated off-target activity. Thus, in addition to facilitating a detailed analysis of the Nep4 subcellular localization in sperm, Nep4::mNG[homo] animals may represent a valuable tool for detailed studies on the molecular function of Nep4 in male fertility. Of note, the analyzed knockdown animals exhibited some residual *nep4* expression in their testes (Supplementary Fig. 3), which needs to be considered when interpreting the associated phenotypes.

### Nep4::mNG exhibits impaired catalytic activity

To understand the molecular effects of the mNeonGreen fusion in more detail, we expressed and purified both a mNeonGreen-tagged and a His-tagged version of Nep4 and analyzed them for their individual catalytic activities. The latter construct has previously been confirmed to be enzymatically active[15,16]. We found that Nep4::His exhibited catalytic activity and specificity as previously described (cleavage between Ala-5 and Val-6 in YLIYAVLa, luminal part of Sarcolamban A[16], Supplementary Fig. 4). By contrast, mNeonGreen-tagged Nep4 did not hydrolyze the applied peptide, indicating that the mNeonGreen fusion resulted in an inactive Nep4

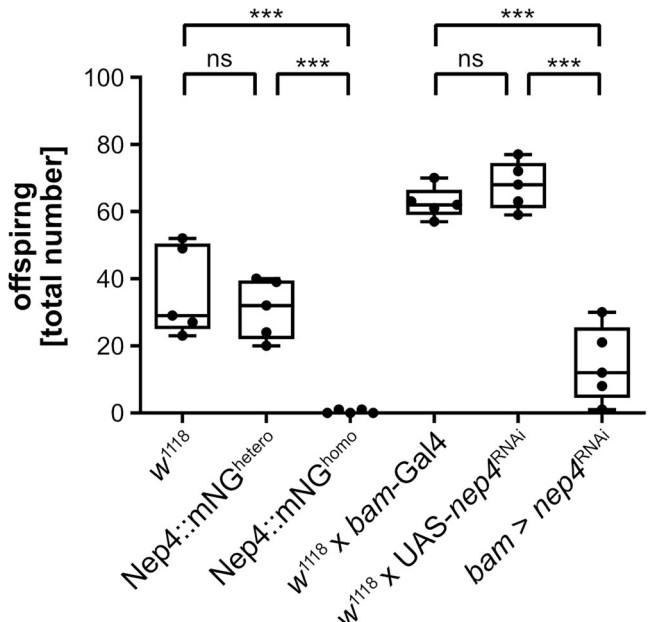

**Fig. 1 | Homozygous Nep4::mNG flies are male sterile.** Male flies of the indicated genotypes were mated to wildtype virgin females. Offspring were counted after 18 days. The fertility of homozygous Nep4::mNG males (Nep4::mNG[homo]) is significantly reduced compared to controls (*w*[1118] or Nep4::mNG[hetero]). Significantly reduced fertility is also evident for *nep4*[RNAi] knockdown males (*bam* > *nep4*[RNAi]), relative to control males (*w*[1118] x *bam*-Gal4 and *w*[1118] x UAS-*nep4*[RNAi]). Asterisks indicate statistically significant differences (***$p < 0.001$, one-way ANOVA followed by Tukey's Multiple Comparison Test; ns = not significant). Each dot represents one individual experiment. Five individual biological replicates were analyzed for each genotype. Due to its temperature dependence, all UAS-Gal4 crossings (*w*[1118] x *bam*-Gal4, *w*[1118] x UAS-*nep4*[RNAi], *bam* > *nep4*[RNAi]) were reared at 27 °C, while the remaining lines (*w*[1118], Nep4::mNG[hetero], Nep4::mNG[homo]) were reared at 22 °C.

protein, or at least an enzyme with abnormal substrate specificity. Of note, neprilysins are characterized by the presence of a distinct sequence motif (CxxW) at their C-terminus. Since this motif is essential to protein folding and maturation[12], fusion of a rather large tag, such as mNeonGreen, at that position apparently impairs the proper function of the enzymes, while the much smaller His-tag is tolerated.

### Nep4 is present in developing spermatids and localizes to the apical tip of individualized sperm

Nep4 is abundantly expressed in adult testes of *Drosophila*[13]. However, a detailed subcellular localization analysis has not been conducted in this tissue yet. We analyzed the generated Nep4::mNG animals accordingly. Significantly, although the Nep4::mNG fusion protein exhibited impaired functionality (Fig. 1, Supplementary Fig. 4), it showed a subcellular localization pattern identical to that of the untagged protein in spermatids and sperm (Supplementary Fig. 5). Thus, the corresponding fly line can be used to track Nep4 localization during spermatogenesis in due detail. The different stages of spermatogenesis are shown schematically in (Fig. 2A). At the apical tip of the testis, germ line stem cells are located. These cells undergo an asymmetrical division resulting in gonioblasts, which are surrounded by two cyst cells. Further mitotic and meiotic divisions lead to the development of 64 round spermatids, which undergo several differentiation steps, including elongation and morphological changes of the nuclei. This phase is referred to as spermiogenesis (Fig. 2A, arrow). Different stages of spermiogenesis can be distinguished by the characteristic shape of the nuclei[1,2], which is visualized by DAPI staining (Fig. 2E). In testes isolated from Nep4::mNG[homo] animals, the Nep4::mNG signal was predominantly visible at the distal tip of developing spermatids in the region towards the basal end (Fig. 2A"). In addition, Nep4::mNG was detected at the tip of individualized

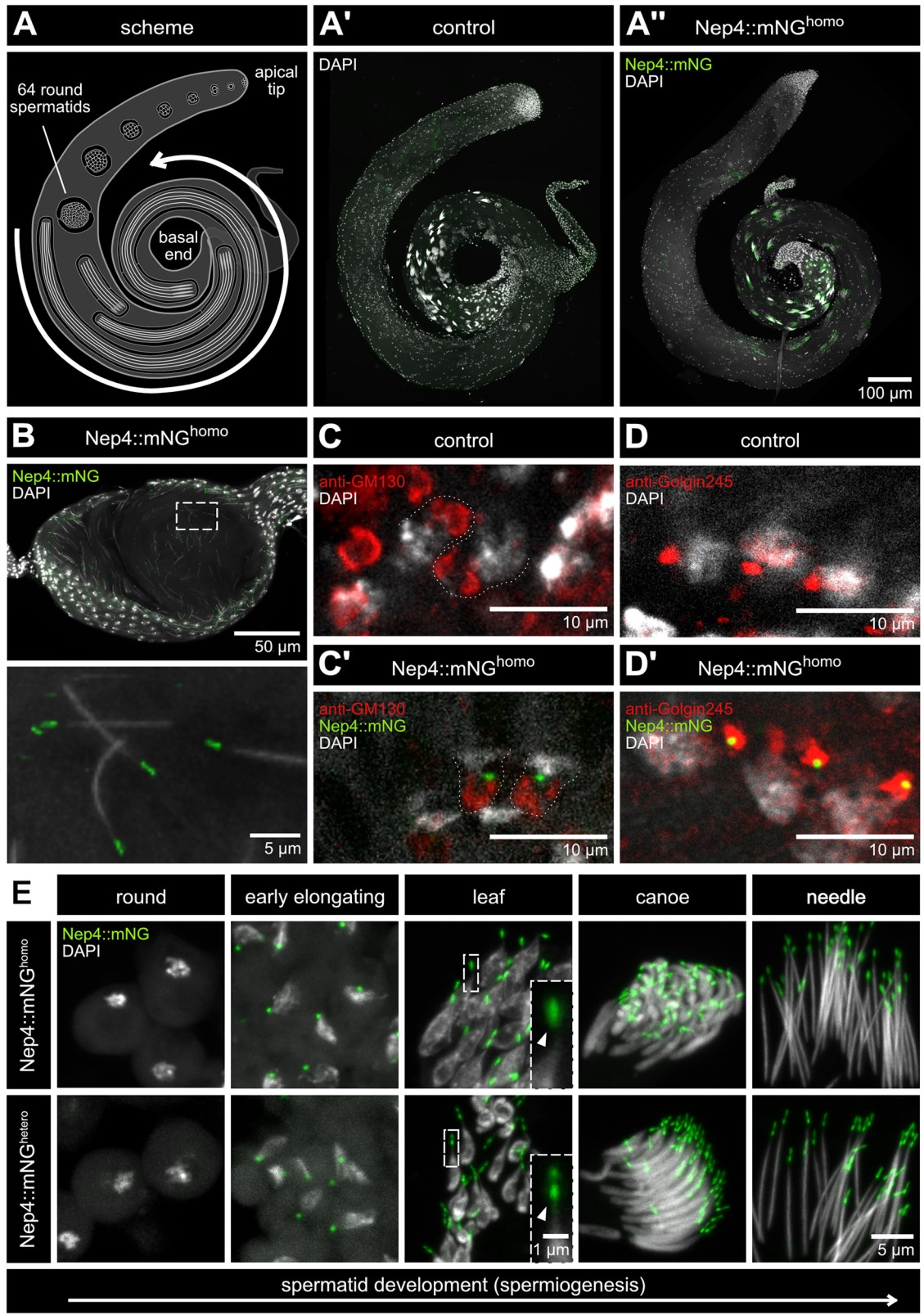

sperm within the seminal vesicle, indicating proper completion of spermatogenesis (Fig. 2B). By contrast, no signal was observed in earlier stages in the apical region of the testis. We also imaged testes isolated from $w^{1118}$ males as a control; no signal above background was detected (Fig. 2A'). At higher magnification, Nep4::mNG was visible in all spermatid stages during spermiogenesis, except for early round spermatids (Fig. 2E). First detection occurred in the form of a distinct spot in early elongating spermatids at the

side of the nucleus where the acroblast is formed. This localization was further specified by co-stainings with the trans-Golgi marker Golgin245 or the cis-Golgi protein GM130, which indicate the aggregation of the Golgi structures near the nucleus to form the acroblast[6]. Co-localization was observed predominantly between Nep4::mNG and Golgin245, indicating Nep4::mNG localization at the trans-Golgi in early elongating spermatids (Fig. 2C, C', D, D'). Throughout development, Nep4::mNG consistently

**Fig. 2 | Nep4::mNG localizes to the apical tip of developing spermatids and individualized sperm.** Testes and seminal vesicles from 1-2 days old males of the indicated genotypes were isolated and stained with DAPI (grey). The genuine mNG signal (green) was preserved during the staining procedure. **A** Different stages of spermatogenesis are depicted in a scheme of the *Drosophila* testis. The phases of development from spermatids to individualized sperm are indicated (arrow). **A', A"** Complete testes isolated from control animals (*w*[1118]) or Nep4::mNG[homo] animals are shown. Stages of spermatogenesis can be deduced from the DAPI staining. In addition to the sperm nuclei, the nuclei of the testis cortex cells are visible. During spermiogenesis, Nep4::mNG localizes to the apical tip of developing spermatids and individualized sperm. No fluorescence signal above background was observed in the green channel of control animals. **B** Overview of a seminal vesicle with the connection to the testis on the left side. In addition to the sperm nuclei, the nuclei of the seminal vesicle cortex cells are visible. Dashed box indicates area of higher magnification, as shown in the image on the right. Nep4::mNG localizes at the sperm tip. **C, C'** The cis-Golgi in early elongating spermatids is stained with anti-GM130

antibodies (red) in testes from control animals (*w*[1118]) and Nep4::mNG[homo] animals. No distinct co-localization is evident (**C'**). Dotted lines indicate the outline of individual structures adjacent to the corresponding nuclei. **D, D'** The trans-Golgi in early elongating spermatids is stained with anti-Golgin245 antibodies in testes from control animals (*w*[1118]) and Nep4::mNG[homo] animals. Nep4::mNG (green) and Golgin245 (red) co-localize in a distinct region of the trans-Golgi (yellow, **D'**). **E** Selected stages of spermiogenesis are shown at higher magnification and sorted by differentiation stage. Developing spermatids from Nep4::mNG[homo] (upper panel) and Nep4::mNG[hetero] animals (lower panel) are depicted. The Nep4::mNG signal is consistently visible adjacent to the nuclei at all stages, except round spermatids. Upon reaching the stage of spermatids with leaf-shaped nuclei, the Nep4::mNG signal appears spherical in Nep4::mNG[homo] animals, while it exhibits a rod-like shape in Nep4::mNG[hetero] animals (arrowheads). Insets depict areas of higher magnification, as indicated by dashed boxes in the overviews, and illustrate the genotype-specific distribution of the Nep4::mNG signal in more detail.

localized close to the nucleus, either in flies homozygous or heterozygous for Nep4::mNG (Fig. 2E). With progressing differentiation, the signal on the distal side of the developing spermatids, which now had leaf-shaped and canoe-shaped nuclei, became increasingly oval. In individualized sperm, Nep4::mNG localized to the apical tip of sperm, clearly anterior to the needle-shaped nuclei. The described spatial and temporal distribution of the Nep4::mNG signal is reminiscent of the pattern of the emerging acrosome, suggesting an acrosomal localization of Nep4 in sperm.

Significantly, the shape of the Nep4::mNG signal exhibited genotype-specific differences in Nep4::mNG[homo] and Nep4::mNG[hetero] animals. These differences became apparent from the stage of developing spermatids with leaf-shaped nuclei (Fig. 2E, insets). While the signal in heterozygous Nep4::mNG animals became increasingly rod-shaped during spermatid differentiation, it remained oval to spherical in homozygous animals (Fig. 2E, arrowheads). Taking into account that Nep4::mNG[hetero] males were fertile and did not exhibit any indication of sperm malformation, we considered the Nep4 localization and the associated signal shape in these animals wildtype.

## Nep4 localizes to the acrosome and is crucial to organelle morphology

To confirm that Nep4 localizes to the acrosome, the subcellular localization of the protein was investigated by correlated light and electron microscopy (CLEM). As shown in Fig. 3, the mNeonGreen-based fluorescence signal was localized to the acrosome region of sperm isolated from homozygous or heterozygous males, with little to no overlap with the closely adjacent nucleus (Fig. 3A, Supplementary Videos 1, 2). For a more detailed evaluation, we analyzed Nep4 localization relative to the acrosomal protein Sneaky (Snky) by employing a Snky::GFP fusion construct[17,18]. In this regard, we combined both fusion constructs, Snky::GFP and Nep4::mNG, in a single fly line. Again, Nep4::mNG[homo] and Nep4::mNG[hetero] animals were analyzed individually, with both lines being homozygous for the Snky::GFP construct. Since the mNG signal was significantly brighter than the GFP signal, the latter was not detected with the corresponding mNG acquisition parameters (Supplementary Fig. 6). Detection of the Snky::GFP fusion protein was instead facilitated by anti-GFP antibody staining. Both proteins, Snky::GFP and Nep4::mNG, localized to the acrosomal region of the sperm, with the Nep4::mNG signal being slightly shifted towards the tip (Fig. 3B). Nevertheless, the signals overlapped in a certain area near the center of the acrosome (Fig. 3B, arrowheads). This pattern was observed in sperm of Snky::GFP; Nep4::mNG[homo] and in Snky::GFP; Nep4::mNG[hetero] animals. However, under homozygous conditions, particularly the Nep4::mNG signal exhibited an altered shape and extension, compared to the heterozygous conditions. In the latter animals, the respective signal was rod-shaped, while in Snky::GFP; Nep4::mNG[homo] animals it appeared rather spherical. The Snky::GFP signal, on the other hand, looked similar under both genetic conditions (Fig. 3B). These results confirm an acrosomal localization of Nep4. Moreover, the altered signal shape in Nep4::mNG[homo]

animals indicates a crucial function of Nep4 in ensuring proper acrosomal morphology.

## Nep4 activity is essential to acrosome integrity and sperm tip structure

To characterize the acrosomal malformations precisely, we quantified the morphological changes at the tips of sperm isolated from the relevant animals (Snky::GFP; Nep4::mNG[hetero] and Snky::GFP; Nep4::mNG[homo]). A fly line expressing the Snky::GFP fusion construct and untagged, endogenous *nep4* served as a control (Snky::GFP) (Fig. 4). In addition to the Snky::GFP and Nep4::mNG signals, we stained corresponding sperm with DAPI (nuclei) and wheat germ agglutinin (WGA) conjugates to further characterize the altered morphology of the acrosomal and the sperm tip region. WGA is a carbohydrate-binding lectin with high affinity to N-acetyl-glucosamine (GlcNAc) and sialic acid moieties that labels the area of the sperm plasma membrane overlaying the acrosome[19]. Using this plasma membrane label as a starting point, we measured signal intensities from the sperm tip towards the nucleus for each channel over a range of 3.5 μm (Fig. 4A, dashed open arrows).

As described above, the Nep4::mNG and the Snky::GFP signals differed in animals heterozygous or homozygous for Nep4::mNG. This effect was reflected in the mean pixel intensity measurements (Fig. 4B-B") and the statistical analyses of the resulting data (Fig. 4C, C'). In Snky::GFP; Nep4::mNG[hetero] animals, both signals showed an asymmetrical distribution along the acrosomal region and, in sum, covered a range of approximately 3 μm from the sperm tip (Fig. 4B'). The Nep4::mNG signal was visible in a bipartite pattern, with two intensity peaks in the anterior acrosomal region. The posterior part of the Nep4::mNG signal exhibited a defined overlap with the Snky::GFP signal. The Snky::GFP signal intensity started to increase at the region of overlap and became clearly visible in the posterior acrosomal region, spanning approximately 2 μm. In addition to these signals, also the WGA signal showed a characteristic pattern, with a distinct peak at the sperm tip that faintly extended up to a distance of about 2 μm towards posterior. Finally, the DAPI signal became detectable at a distance of approximately 2 μm from the sperm tip and continuously increased towards the posterior part of the sperm. Of note, the described distribution of the Snky::GFP, WGA, and DAPI signals in Snky::GFP; Nep4::mNG[hetero] animals was identical to the distribution of the corresponding signals in control animals, which expressed the Snky::GFP construct and wildtype *nep4* (Fig. 4B). Statistical analyses of the corresponding signal intensities confirmed this indication (Fig. 4C-C'"). For these analyses, the average distances of the maximum intensities of the Nep4::mNG, the Snky::GFP, and the WGA signals from the sperm tip were assessed. In addition, the distance from the sperm tip at which the DAPI signals surpassed a predefined lower threshold was determined. The corresponding results were statistically indifferent for Snky::GFP and Snky::GFP; Nep4::mNG[hetero] animals. Of note, in addition to the high similarity between Snky::GFP and Snky::GFP; Nep4::mNG[hetero] animals, a comparable signal distribution was also

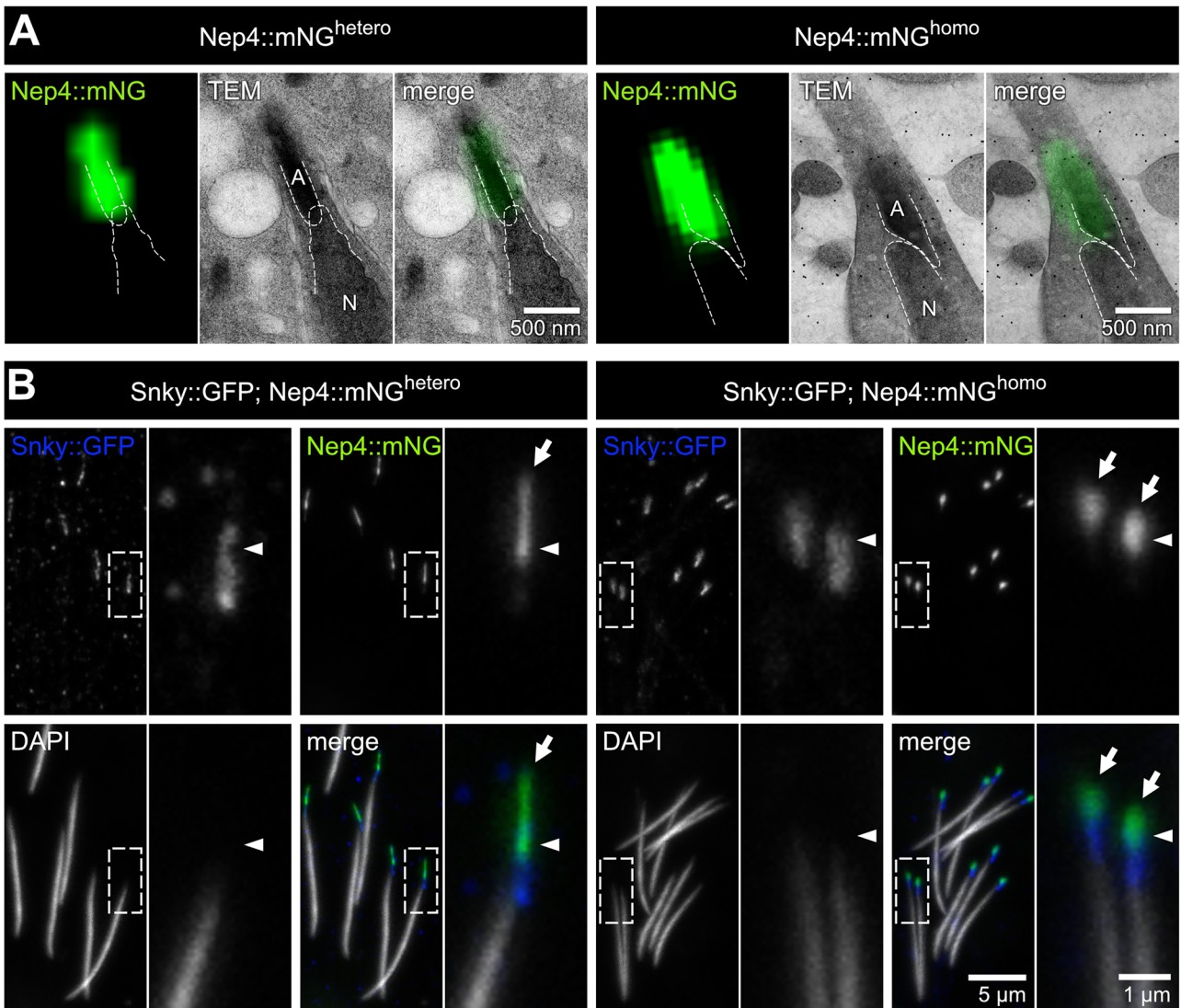

**Fig. 3 | Nep4::mNG localizes to the acrosome and affects organelle morphology.** Sperm from 1–2 days old flies were dissected and analyzed for Nep4::mNG localization. **A** Correlated light and electron microscopy (CLEM) analysis confirms that the Nep4::mNG-based fluorescence signal localizes to the acrosome region of sperm isolated from heterozygous (left panel) or homozygous (right panel) males. Dashed lines indicate organelle boundaries. A: acrosome; N: nucleus. **B** Sperm expressing Snky::GFP and Nep4::mNG were stained with anti-GFP antibodies (blue) and DAPI (grey). The genuine Nep4::mNG signal (green) was preserved during fixation. Single channel and merged images are depicted. Dashed boxes indicate areas of higher magnification, as shown to the right of the respective overview image. Sperm of Snky::GFP; Nep4::mNG^hetero (left panel) and Snky::GFP; Nep4::mNG^homo (right panel) animals are analyzed. Snky::GFP and Nep4::mNG localize to the acrosomal region of the sperm tip and overlap at a defined region in the center of the acrosome (arrowheads). In the sperm of heterozygous Nep4::mNG animals, both signals exhibit a rod-shaped form. Under homozygous conditions, the Nep4::mNG signal appears more spherical (arrows).

observed in animals that did not express the Snky::GFP fusion construct (Supplementary Fig. 7). Thus, the Nep4 localization was not affected by the Snky::GFP transgene. These results confirm a wildtype localization pattern of the measured signals in animals heterozygous for Nep4::mNG. Consequently, sperm isolated from Snky::GFP; Nep4::mNG^hetero animals are considered a valid control.

In contrast to all controls, the corresponding signals in Snky::GFP; Nep4::mNG^homo animals exhibited a much stronger overlap and did not show the distinct partitioning. The specific intensities of the Snky::GFP signal, the Nep4::mNG signal, and the WGA signal rather merged into one combined peak (Fig. 4B"). Although the Nep4::mNG signal was still slightly shifted towards the sperm tip compared to the Snky::GFP signal, both signals showed a significant overlap in the homozygous animals that was not observed in the heterozygous controls. Additionally, both signals combined covered a shorter distance of approximately 2 µm from the sperm tip in Snky::GFP; Nep4::mNG^homo animals, while this region comprised approximately 3 µm in Snky::GFP; Nep4::mNG^hetero animals.

The reduction in length was mainly due to a much narrower distribution of the Nep4::mNG signal, while the Snky::GFP signal remained largely unchanged. These differences were also evident in the statistical analyses (Fig. 4C-C'). The maximum intensities of Snky::GFP and Nep4::mNG were significantly shifted towards anterior in Snky::GFP; Nep4::mNG^homo animals, compared to the controls. The altered signal distribution of Nep4::mNG and Snky::GFP in the sperm of homozygous animals suggests an abnormal morphology of the acrosome in these animals, which is caused by impaired Nep4 functionality. Compromised acrosomal morphology was further supported by the WGA and DAPI signals in the sperm of homozygous Nep4::mNG animals. The WGA signal showed a broader and more even distribution (Fig. 4B"), with the maximum signal intensity being significantly shifted towards posterior, relative to the controls (Fig. 4C"). This indicates a change in the plasma membrane structure covering the acrosomal region. Moreover, the DAPI signal intensity reached a linear increase significantly earlier in Snky::GFP; Nep4::mNG^homo animals compared to control animals (Fig. 4C'''),

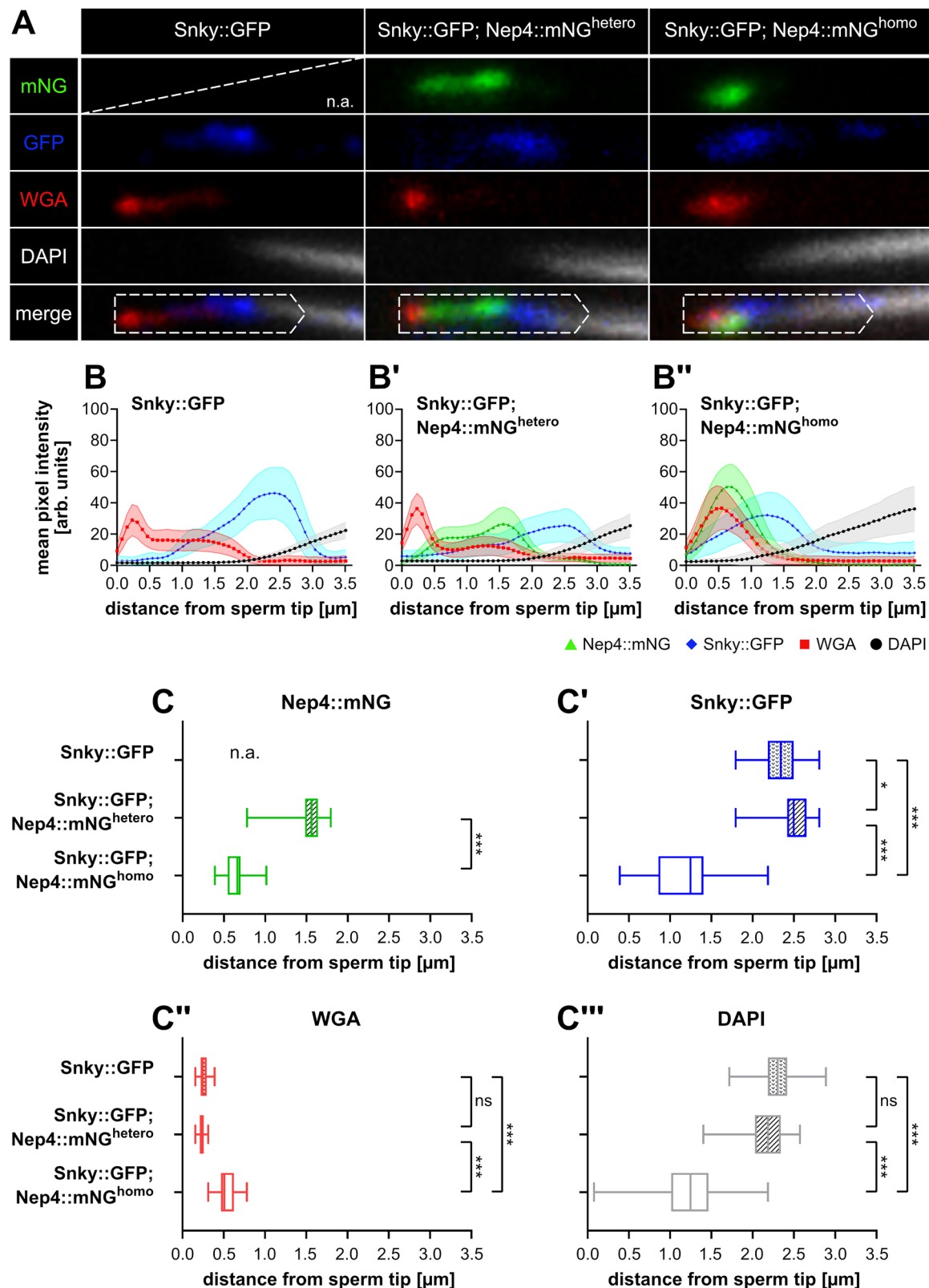

reflecting a more anterior localization of the nucleus and thus a considerably condensed acrosomal region.

In sum, the distribution of all four factors analyzed in animals with functional Nep4 was highly structured and exhibited a clear and distinct pattern. By contrast, sperm of flies homozygous for the Nep4::mNG fusion construct exhibited a severely impaired sperm tip partitioning that did not show the characteristic separate signal distribution for Nep4::mNG, Snky::GFP, and WGA. Rather, they were much more evenly distributed and had wider areas of overlap. In addition, the nucleus was located more anteriorly, indicating a somewhat collapsed acrosomal area with reduced suborganellar patterning. These data indicate that functional Nep4 is crucial for proper acrosomal morphology and, consequently, structure of the sperm tip.

**Fig. 4 | Homozygous conditions for Nep4::mNG affect sperm tip structure.** Sperm from 1–2 days old flies of the indicated genotypes were dissected and stained with anti-GFP antibodies (blue), WGA (red), and DAPI (grey). The genuine Nep4::mNG signal (green) was preserved during the staining procedure. Sperm of Snky::GFP (control), Snky::GFP; Nep4::mNG^hetero, and Snky::GFP; Nep4::mNG^homo animals are analyzed. **A** The distal part of the sperm tip is depicted in single channel images and a merged image for each genotype. Dashed open arrows indicate regions (0.7 ×3.5 μm) that were analyzed by pixel intensity measurements. **B–B"** The mean values of the individual pixel intensities ± SD are shown. For each genotype, at least 50 individual sperm cells isolated from the testes of at least five different animals were analyzed.

While all signals exhibit a well distinguishable distribution in Snky::GFP and Snky::GFP; Nep4::mNG^hetero animals, this characteristic pattern is lost in Snky::GFP; Nep4::mNG^homo animals. **C–C"'** Characteristic parameters of the pixel intensity measurements are analyzed. The individual distances of the maximum intensities of the Nep4::mNG (**C**), the Snky::GFP (**C'**), and the WGA signals (**C"**) from the sperm tip are displayed. In addition, the distance from the sperm tip at which the DAPI signal intensities surpassed a predefined threshold are shown (**C"'**). Asterisks indicate statistically significant deviations from controls (Snky::GFP; *$p < 0.05$, ***$p < 0.001$; one-way ANOVA followed by Tukey's Multiple Comparison Test; ns = not significant). n.a. = not applicable.

To verify that the described effects on the acrosomal morphology were indeed caused by impaired Nep4 function and not by CRISPR/Cas9-mediated off-target activity, we analogously analyzed the subcellular localization of relevant factors, including WGA and DAPI, in nep4^RNAi knockdown animals. Germline-specific knockdown was achieved by using bamGal4 as a driver (Fig. 5). In the control lines ($w^{1118}$ x bam-Gal4; $w^{1118}$ x UAS-nep4^RNAi), the signal distribution of WGA and DAPI was largely identical to the distribution in Snky::GFP and Snky::GFP; Nep4::mNG^hetero animals (Fig. 5B, B'), thus resembling the wildtype situation. By contrast, the WGA and the DAPI signal distributions in nep4^RNAi knockdown animals (bam > nep4^RNAi, Fig. 5B") were highly reminiscent of the respective distribution in the Snky::GFP; Nep4::mNG^homo animals. In detail, the WGA signal was more evenly spread and exhibited a broader dominant signal, with the maximum signal intensity being significantly shifted towards posterior (Fig. 5C). Moreover, the DAPI signal intensity surpassed the predefined threshold earlier in nep4^RNAi animals than in the controls (Fig. 5C'). While the former was indicative of an impaired plasma membrane structure at the sperm tip, the latter reflected a more anteriorly localized nucleus in the sperm of knockdown animals, likely accompanied by a reduced or morphologically abnormal acrosomal area. Both effects were highly reminiscent of the pattern in Snky::GFP; Nep4::mNG^homo animals. The consistency of the nep4^RNAi knockdown data with the effects observed in Snky::GFP; Nep4::mNG^homo animals provides further evidence for a crucial function of Nep4 in ensuring proper acrosomal morphology and sperm tip partitioning. Of note, we also measured the lengths of the sperm nuclei of all genotypes analyzed in this study (Supplementary Fig. 7D, D'). No significant differences were found, suggesting a similar stage of development of the analyzed sperm.

### Impaired Nep4 function affects acrosomal size and morphology at the ultrastructural level

We applied serial block-face scanning electron microscopy (SBF-SEM) to quantify the effects of impaired Nep4 function on acrosome integrity at the ultrastructural level (Supplementary Fig. 8). The 3D reconstructions were used to calculate the acrosome length, width, and volume in sperm of animals of relevant genotypes (Fig. 6; Supplementary Fig. 9; Supplementary Videos 3, 4, 5). In control animals ($w^{1118}$ x UAS-nep4^RNAi) the acrosome length comprised 2.98 μm on average, while it was significantly shorter in Nep4::mNG^homo and nep4^RNAi knockdown animals, exhibiting average lengths of 1.98 μm and 1.84 μm, respectively (Fig. 6B). These measurements substantiate our data from the pixel intensity analyses (Figs. 4, 5). In those experiments, the acrosomal regions of the sperm tip, which were in sum covered by the Nep4::mNG and the Snky::GFP signals, were approximately 3 μm in control animals and approximately 2 μm in homozygous Nep4::mNG animals. In addition to the length, the SBF-SEM dataset also enabled the calculation of the acrosome volume. Compared to control animals (0.18 μm³), the volume in Nep4::mNG^homo (0.12 μm³) as well as in nep4^RNAi animals (0.13 μm³) was significantly reduced (Fig. 6D). Interestingly, the detailed evaluation of the individual 3D reconstructions also confirmed an abnormal morphology of acrosomes in sperm from both Nep4::mNG^homo and nep4^RNAi animals (Fig. 6A, insets; Supplementary

Fig. 9). Prominent alterations included a significantly increased diameter of corresponding acrosomes, compared to control organelles (Fig. 6C). In the latter, the widest diameter measured 0.44 μm, whereas it was 0.50 μm in Nep4::mNG^homo animals and 0.56 μm in nep4^RNAi animals. However, acrosomes of all genotypes still exhibited a flattened area in the posterior region of the individual organelles, which reflects the position of the directly adjacent nucleus. This particular region appeared to be not affected by the morphological impairments that were obvious in the anterior part of the acrosome (Fig. 6A, lower panel; Supplementary Fig. 9). Overall, the 3D reconstructions confirmed that sperm isolated from animals with impaired Nep4 activity contained acrosomes with severely impaired morphology. Moreover, the dataset revealed a significant reduction in the length and volume as well as an increase in the diameter of the corresponding acrosomes.

### Impaired Nep4 function affects sperm storage, fertilization, and initiation of embryonic development

To gain further mechanistic insight into how the observed abnormalities at the acrosomes lead to male sterility, we analyzed the seminal receptacles and the laid eggs of wildtype females mated with Nep4::mNG^homo or Nep4::mNG^hetero males for the presence of corresponding sperm. This way, we aimed to differentiate between abnormal spermatogenesis, abnormal fertilization, or failure to initiate embryonic development as potential underlying causes. As a result, we found that the sperm of Nep4::mNG^homo males were present in the seminal receptacles, but were discarded from the female storage organs rather quickly. Relative to the heterozygous control, only 54% of sperm from Nep4::mNG^homo males were present in the seminal receptacle of corresponding females 8 h after the end of copulation (AEC). The same effect was observed 24 h AEC, albeit to a slightly lower extent (60%, Fig. 7A, B). We furthermore analyzed eggs laid by wildtype females that had copulated with Nep4::mNG^homo or Nep4::mNG^hetero males (Fig. 7C-E). For the control (Nep4::mNG^hetero) almost every analyzed egg initiated embryonic development (Fig. 7E). In contrast, none of the eggs laid by females that had copulated with Nep4::mNG^homo males exhibited embryonic development. Furthermore, 63% of the corresponding eggs were found to be devoid of a sperm nucleus, while 31% of these eggs contained a condensed, needle-shaped sperm nucleus. Interestingly, a further 6% of the eggs contained a condensed sperm nucleus with adjacent Nep4::mNG signal, suggesting presence of an intact acrosome (Fig. 7E).

These results indicate that impaired Nep4 function affects male fertility at multiple levels, but predominantly following sperm transfer to females. In addition to the sperm storage time in seminal receptacles, the fertilization efficiency and the ability of corresponding sperm to initiate embryogenesis were impaired. Our data are consistent with previous experiments on nep4 mutants that lack most of the extracellular protein domain, and thus catalytic activity[11].

### Discussion

In previous studies, nep4 was identified in a screen for genes essential to male fertility[10]. In addition, work on deletion mutants lacking most of the extracellular Nep4 protein domain suggested that male sterility results from defects following sperm transfer to the female reproductive tract.

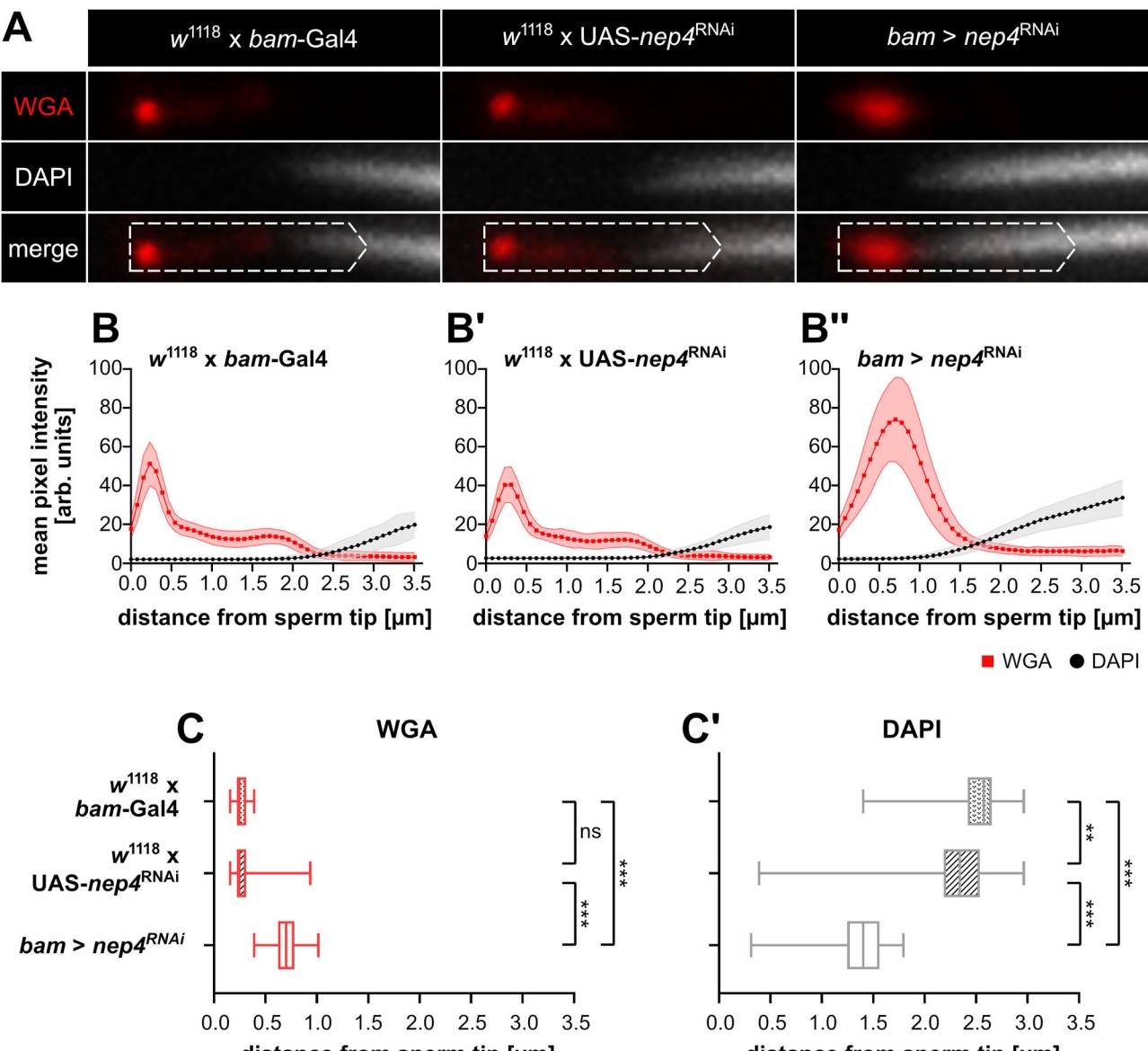

**Fig. 5 | Reduced Nep4 levels affect sperm tip structure.** Sperm from 1–2 days old flies of the indicated genotypes were dissected and stained with WGA (red) and DAPI (grey). Sperm of *nep4*^RNAi knockdown animals (*bam > nep4*^RNAi) and corresponding controls (*w*^1118 x *bam*-Gal4; *w*^1118 x UAS-*nep4*^RNAi) were analyzed. **A** The distal part of the sperm tip is depicted in single channel images and a merged image for each genotype. Dashed open arrows indicate regions (0.7 ×3.5 μm) analyzed by pixel intensity measurements. **B–B''** The mean values of the individual pixel intensities ± SD are shown. For each genotype, at least 50 individual sperm cells isolated from the testes of at least five different animals were analyzed. While the sperm of control animals show a well-confined distribution pattern for the WGA signal, the distribution at the sperm tip of *nep4*^RNAi knockdown animals is much broader and the DAPI signal is shifted towards anterior. **C–C'** Characteristic parameters of the pixel intensity measurements are analyzed. **C** Distance of the maximum intensities of the WGA signals from the sperm tip. **C'** Distance from the sperm tip at which the DAPI signal intensities surpassed a predefined threshold. Asterisks indicate statistically significant deviations from controls (**$p < 0.01$, ***$p < 0.001$; one-way ANOVA followed by Tukey's Multiple Comparison Test; ns = not significant).

Incomplete sperm maturation was proposed to cause impaired sperm storage in female storage organs, reduced fertilization efficiency, and failure to initiate embryonic development[11]. However, no detailed morphological analyses were performed in sperm or testes of corresponding mutants. In addition, the subcellular localization of Nep4 in these tissues as well as its specific molecular function in ensuring male fertility remained elusive. Interestingly, *Nep2*-deficient mice also exhibit reduced fertility and impaired early embryonic development, similar to the phenotypes observed in the *Drosophila nep4* mutants. Although the underlying molecular mechanism remained unclear, impaired sperm maturation was suggested as the cause of these phenotypes[14].

To address these issues and to further investigate the relevance of neprilysins in ensuring male fertility, we analyzed the subcellular localization and function of Nep4 during *Drosophila* spermatogenesis. We found that Nep4 localizes to the acrosome in sperm and that activity of the peptidase is crucial for proper acrosome morphology and sperm tip structure (Figs. 2–6; Supplementary Fig. 9; Supplementary Videos 3, 4, 5).

In line with murine *Nep2*, which was detected in round and elongated spermatids[20], we found *Drosophila* Nep4 in nearly all spermatid stages during spermiogenesis (Fig. 2). It was consistently present in close proximity to the nucleus, closely resembling the developing acrosome. In mature sperm, Nep4 localized to the anterior region of the acrosome, as confirmed by CLEM analyses and co-staining of the acrosomal protein Snky[18] (Fig. 3). This distinct subcellular localization pattern of Nep4 indicates a function during spermiogenesis, and, more specifically, during acrosome development.

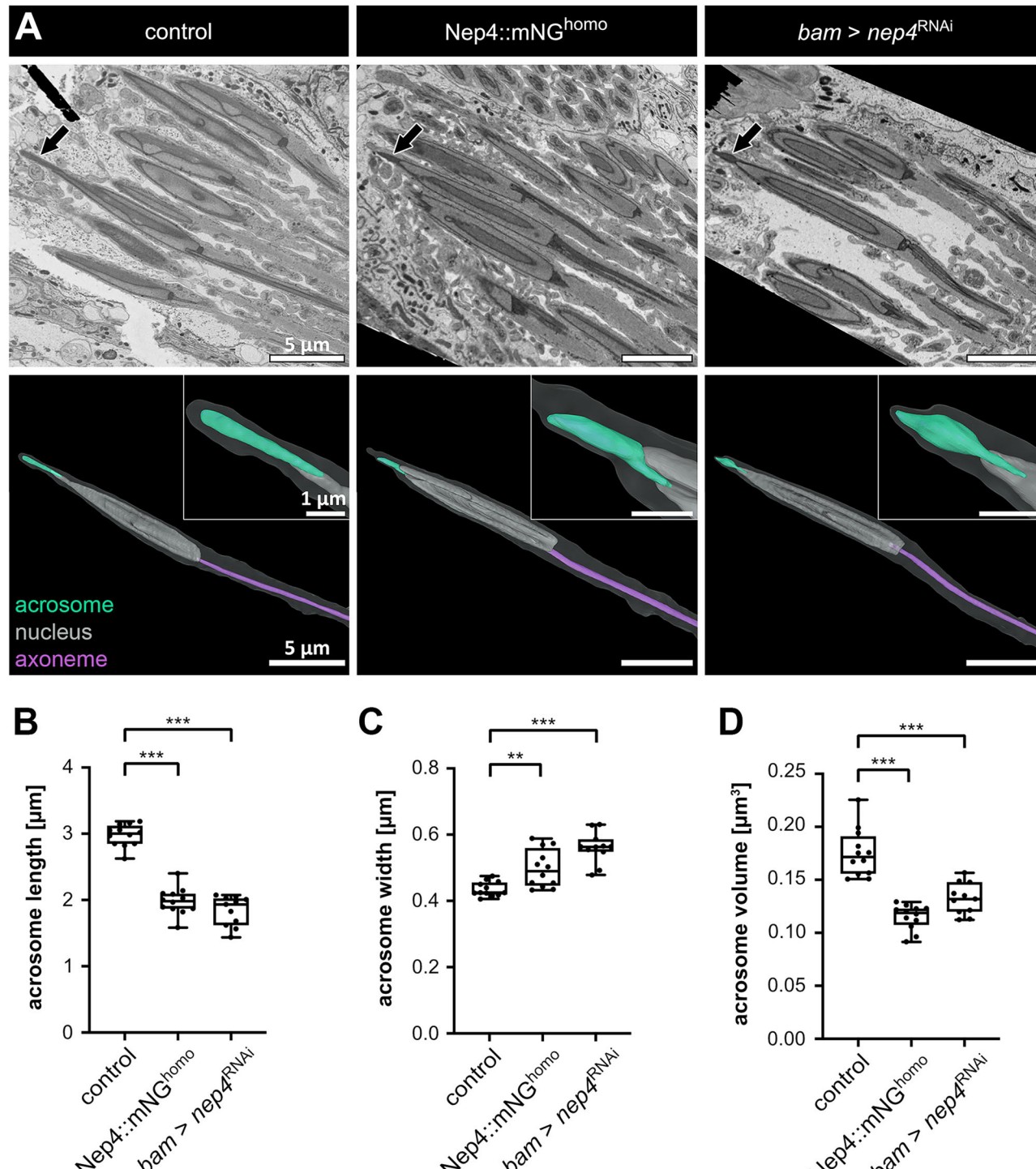

**Fig. 6 | Impaired Nep4 activity affects acrosome structure.** Testes from 1–2 days old flies of the indicated genotypes were analyzed by serial block-face scanning electron microscopy (SBF-SEM). **A** EM images show an optical longitudinal section of a representative single sperm tip for each genotype. The acrosome is marked by arrows (upper panel). In addition, volume-rendered images of the respective sperm tips are shown (lower panel). In the 3D model, the acrosome is depicted in turquoise, the nucleus in grey, and part of the axoneme in purple. The shadow around the structures outlines the sperm cell. Insets depict the corresponding acrosomes at higher magnifications (for the insets, the scale bar always corresponds to 1 μm). **B–D** Three-dimensional reconstructions were used to measure acrosome length, width, and volume. All factors are significantly affected in sperm isolated from Nep4::mNG^homo and nep4^RNAi knockdown animals, relative to sperm isolated from control animals ($w^{1118}$ x UAS-nep4^RNAi). Asterisks indicate statistically significant deviations (**$p < 0.01$, ***$p < 0.001$; one-way ANOVA followed by Tukey's Multiple Comparison Test). Each dot represents one individually analyzed acrosome.

Significantly, we found that the Nep4 signal distribution was different in sperm isolated from animals that were either homozygous or heterozygous for a Nep4::mNG genomic insertion (Figs. 2–4). Based on the fact that heterozygous Nep4::mNG males were fertile and the subcellular

distribution of analyzed factors in corresponding sperm was largely identical to endogenous controls, we considered the rod-shaped Nep4::mNG localization in sperm from heterozygous animals wildtype and used these flies as an additional control. Consequently, the altered and more spheroid

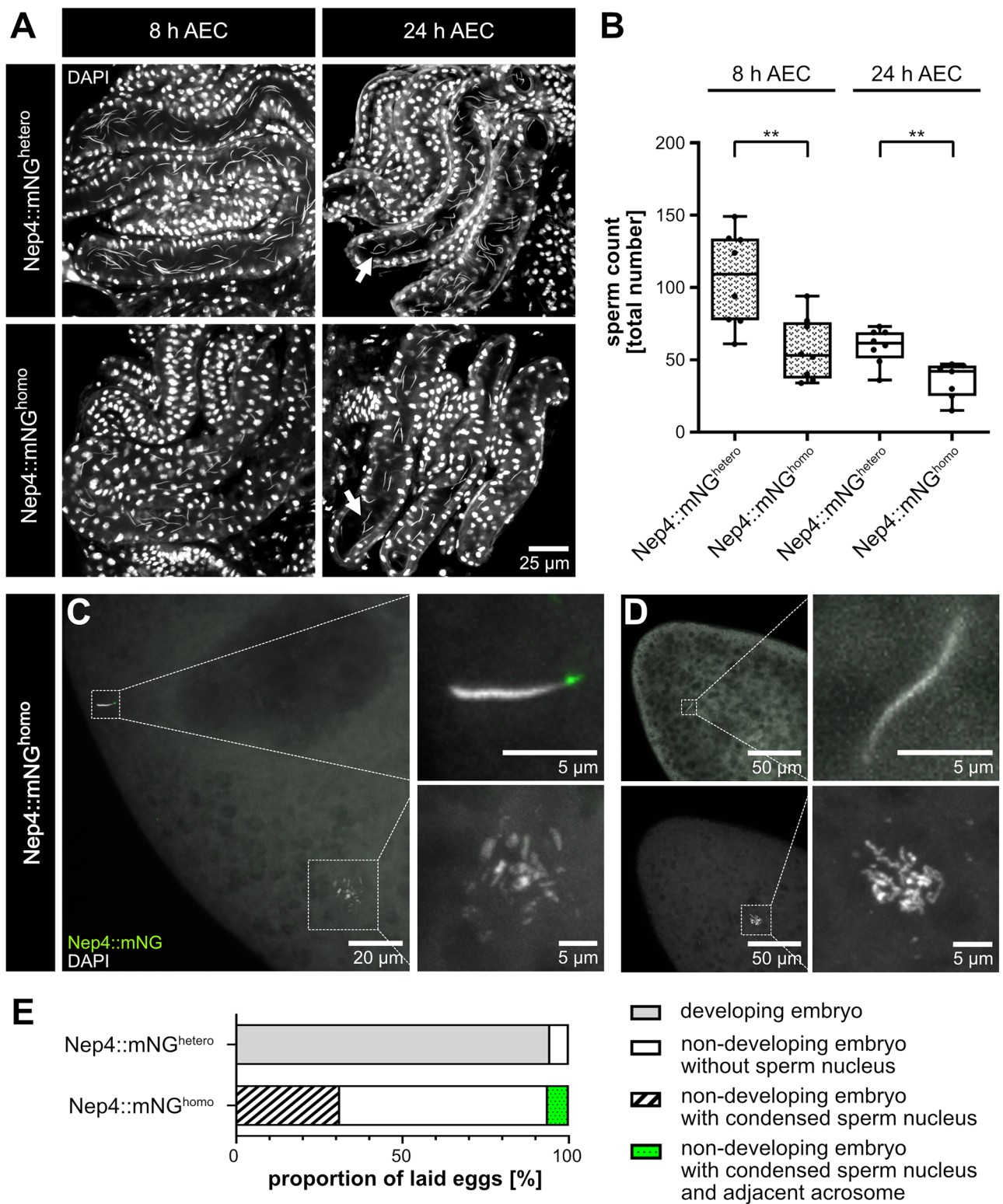

Nep4::mNG signal shape in sperm from homozygous animals likely represents the relevant phenotype that eventually causes male sterility in these animals.

In addition to Nep4, the subcellular distribution of other factors was also compromised in sperm from animals with impaired Nep4 function. These factors included the acrosomal protein Snky, sperm surface glyco-proteins, and the sperm nuclei (Figs. 4, 5). In contrast to the structured distribution pattern observed in controls, the corresponding signals from homozygous Nep4::mNG and from *nep4* knockdown animals exhibited significantly stronger overlap with reduced partitioning and more anteriorly localized sperm nuclei. The observed morphological changes at the sperm tip are somewhat similar to the pathology of globozoospermia, a human disorder of sperm morphology mainly characterized by absence of acrosomes in round-headed mature sperm[21–23]. However, the effects were less pronounced in the flies, as the acrosomes were not absent, but merely deformed. Furthermore, the characteristic round-shaped sperm heads,

**Fig. 7 | Impaired Nep4 activity affects sperm storage, fertilization, and initiation of embryonic development. A** Seminal receptacles from wildtype females that had copulated with males of the indicated genotypes were collected at specific time points after the end of copulation (AEC) and stained with DAPI to label nuclei (grey). The sperm nuclei were identified and counted on the basis of their characteristic elongated shape (arrows). **B** Quantification of (**A**). The number of sperm in seminal receptacles of females that had copulated with Nep4::mNG[homo] males is significantly lower than the corresponding number for the heterozygous control. The difference is significant for both 8 h AEC and 24 h AEC (**p < 0.01, unpaired t-test, two-tailed). **C–D** Eggs laid by wildtype females that had copulated with Nep4::mNG[hetero] or Nep4::mNG[homo] males were collected and stained with DAPI to visualize maternal and paternal DNA. The Nep4::mNG signal was preserved during the staining procedure. **C** Representative egg. The Nep4::mNG signal indicates presence and localization of the acrosome that appears to be attached to the needle-shaped sperm nucleus (right panel, upper image). The maternal DNA appears in a prometaphase composition (right panel, lower image). **D** Representative egg in two different focal planes to visualize both maternal and paternal DNA. While maternal DNA (lower panel) as well as the needle-shaped sperm nucleus (upper panel) is clearly visible, no Nep4::mNG signal is present. **E** Quantification of (**C–D**). While for the control (Nep4::mNG[hetero]) almost every analyzed egg initiated embryonic development, none of the eggs laid by females that had copulated with Nep4::mNG[homo] males initiated development. Moreover, 63% of corresponding eggs did not contain a sperm nucleus, 31% contained a condensed sperm nucleus, and 6% contained a condensed sperm nucleus and a Nep4::mNG signal.

---

which mainly result from round nuclei in human globozoospermia, were not observed in the affected flies. Nevertheless, the detrimental effects in *Drosophila* were so severe that they lead to almost complete infertility (Fig. 1).

To characterize the mechanistic basis of the impaired sperm tip morphology and the concomitant sterility in more detail, we analyzed the affected distribution of the sperm tip specific factors more thoroughly. Interestingly, in both affected and unaffected acrosomes, Snky localized more posteriorly than Nep4 with overlaps being evident only in the central region of the organelle (Figs. 3, 4). This pattern suggests presence of subcompartments within the acrosomal membrane, as already reported for mammalian acrosomes. Here, the inner acrosomal membrane (IAM) differs from the outer acrosomal membrane (OAM) by containing proteins that attach the acrosome to the adjacent nucleus via a structure referred to as the acroplaxome[9,24]. In addition, the equatorial segment is a specialized region of the mammalian acrosome membrane. It is located between the IAM and OAM and plays a crucial role in fertilization[25,26]. Significantly, we found that compartmentalization of the acrosomal membrane was less pronounced in response to impaired Nep4 activity. Especially the Nep4 signal appeared more spherical in affected acrosomes and exhibited a broader overlap with the Snky signal (Figs. 3, 4). This result suggests a critical relevance for Nep4 in establishing or maintaining the suborganellar structure of the acrosome. Interestingly, the very posterior part of the individual acrosomes, marked by the presence of Snky, appeared largely intact as it was somewhat flattened in both control and affected sperm (Supplementary Fig. 9). This flattened area defines a position with immediate vicinity to the adjacent sperm nucleus[27]. Considering the fact that Snky contains protein interaction sites that were suggested to contribute to the organization and maintenance of macromolecular complexes at the acrosomal membrane[18], corresponding interactions may link the posterior part of the acrosome to the nucleus. This link seems to be unaffected in animals with impaired Nep4 function. As mentioned above, the acroplaxome, which anchors the acrosome to the nucleus, has a comparable function in mammals[24]. These data suggest that reduced Nep4 abundance predominantly affects the middle and anterior regions of the acrosome, which is in accordance with the localization pattern of Nep4 (Figs. 3, 4; Supplementary Fig. 7).

Given the extent of the morphological changes in the acrosome, disturbed interactions between the acrosome and the cytoskeleton could be involved in the development of the phenotype. To assess this possibility, we did a series of pull-down assays using sperm-specific expression of a GFP-tagged Nep4 construct as bait. Applicability of this construct for pull-down assays has been confirmed previously[16]. We did not identify any cytoskeletal component interacting with Nep4 (Supplementary Fig. 10; Supplementary Data 2). Moreover, gene ontology analyses of the few factors that co-precipitated with Nep4 (Supplementary Data 2) did not yield any significantly affected biological processes or molecular functions (ShinyGO v0.81; FDR cutoff 0.05). Thus, impaired catalytic activity of the enzyme rather than a disturbed protein-protein interaction appears to be the sole mechanistic basis for the observed phenotypes. Abnormal catalytic activity of the mNeonGreen-tagged Nep4 construct was experimentally confirmed (Supplementary Fig. 4). However, the identity of the relevant peptides that accumulate in response to impaired Nep4 activity remains to be determined. Given the morphological defects that were obvious in acrosomes with dysfunctional Nep4, the corresponding peptide substrates, rather than Nep4 itself, may be involved in acrosomal-cytoskeleton interactions. Their accumulation would then result in ectopic interactions, which ultimately trigger the morphological defects. Of note, the fact that membrane-bound peptides are released from the membrane as a result of Nep4-mediated hydrolysis has already been confirmed, e.g., for the SERCA-regulatory Sarcolamban peptides[16]. On the other hand, the observed deformations can also be interpreted as organelle swelling, which may indicate a disturbed osmotic regulation within the acrosome. In this scenario, the accumulation of soluble peptides within the acrosome would lead to an influx of water and ultimately to swelling of the organelles.

As a result of the acrosomal malformation, the distribution of glycoproteins on the surface of the sperm of animals with impaired Nep4 function was also altered, as confirmed by WGA stainings (Figs. 4, 5). In control flies, these glycoproteins are distinctly localized to the sperm tip. By contrast, in animals with impaired Nep4 function, they were much more widely distributed in the plasma membrane of the sperm. These changes in the sperm glycocalyx, a coat of glycoconjugates surrounding the cells, likely contribute to the impaired fertility of affected males. In mammals and *Drosophila*, the sperm glycocalyx is essential for the survival of the sperm in the female tract and for oocyte fertilization[4,28–30]. Interestingly, we found that, in comparison with the heterozygous control, there was a significantly reduced number of Nep4::mNG[homo] sperm in the seminal receptacle of correspondingly mated females, which may be a direct consequence of an abnormal sperm glycocalyx (Fig. 7). Furthermore, the lower number of fertilized eggs that we observed for Nep4::mNG[homo] sperm may also be due to an altered glycocalyx, as the sperm-egg interaction, especially at the micropyle, may be affected. However, even when the Nep4::mNG[homo] sperm entered the eggs, embryonic development was not initiated and the sperm nuclei remained in a condensed, needle-shaped conformation. In this context, our data suggest that at least the breakdown of the sperm plasma membrane is unaffected, as the majority of the fertilized eggs did not exhibit a Nep4::mNG signal in proximity to the condensed nucleus. However, further studies are required to determine which specific process is affected by Nep4 following sperm entry into the egg.

In conclusion, we found that Nep4 functionality is crucial to proper acrosome and sperm tip structure, which eventually ensures male fertility. Compromised catalytic activity of Nep4 results in morphological and structural impairments of the acrosome and, consequently, of the sperm tip. These malformations impair male fertility at multiple levels, with the strongest effects being observed after the transfer of corresponding sperm to the females. In particular, the storage time of the sperm in the seminal receptacle as well as the ability of the sperm to fertilize eggs and initiate embryogenesis are compromised. While further investigation is required to assess the underlying molecular mechanisms in detail, particularly regarding phenotype-relevant Nep4 peptide substrates, this work extends the current knowledge on the physiological relevance of neprilysins. Given the

high functional conservation of the identified factors, the data generated herein may be significant also for studies investigating the fertility of vertebrates, including humans, in health and disease.

## Methods

### Fly strains

The following *Drosophila* lines were used in this work. $w^{1118}$ was used as a control strain and *bam*-Gal4 (RRID:BDSC_80579) as a driver line. Knockdown of *nep4* was achieved using line 100189 (KK library, Vienna *Drosophila* Resource Center [VDRC], RRID:Flybase_FBst0472063). High *nep4*-specific knockdown efficiency of the respective hairpin was confirmed previously[15,16,31]. Efficient knockdown in testes was reassessed and confirmed in this study (Supplementary Fig. 3). A second *nep4*-specific RNAi construct (line 16669, GD library, VDRC) did not significantly reduce *nep4* transcript levels[31]. This line was excluded from further analysis. The Snky::GFP line was a kind gift from Dr. Barbara T. Wakimoto (University of Washington, WA, USA[18];).

### Generation of the transgenic Nep4::mNeonGreen fly line

Nep4::mNeonGreen flies (Nep4::mNG) were generated by CRISPR/Cas9-assisted homology directed repair to insert a C-terminal in-frame mNeonGreen tag into the endogenous locus of *nep4*. The experimental work was done at WellGenetics Inc. (New Taipei City, Taiwan (R.O.C.)). Using one guide RNA (TGCAGCGTTTGGTAGTCGGA[GGG], no predicted off-targets, WellGenetics Inc.) and a dsDNA donor plasmid, the stop codon of *nep4* was replaced by a knock-in cassette (XbaI_mNeonGreen_LoxP_3xP3-GFP_LoxP). The XbaI restriction site encodes a serine and an arginine, serving as a spacer between Nep4 and mNG. 3xP3-GFP was used as a selection marker and was excised by LoxP/Cre recombination in a second step. This resulted in a fly line with endogenous expression of a Nep4::mNG fusion protein. This strain is further referred to as Nep4::mNG$^{homo}$ for homozygous conditions and Nep4::mNG$^{hetero}$ for heterozygous animals.

### Fertility assay

Male fertility was assessed according to ref. [17]. In brief, 3 days old single males were mated to two virgin wildtype females in vials for 7 days. Afterwards, flies were removed from vials and the number of offspring was counted on day 18. The assay was performed at room temperature (RT) for Nep4::mNG animals and respective controls. The fertility assay for *bam*-driven *nep4*$^{RNAi}$ knockdown, including corresponding controls, was performed at 27 °C. For all genotypes, a minimum of five individual biological replicates was analyzed.

The assessment of female fertility was performed similarly, with the modification that only single-pair crosses were used. Here, one 3 days old wildtype male was mated to one virgin female per replicate.

### Sperm count and fertilization rate estimation

Sperm in female storage organs were counted as described in ref. [32]. To assess sperm entry and fertilization, wildtype females that had copulated with males of the analyzed genotypes were transferred to egg collection vials. After 24 h, laid eggs were collected and stained with DAPI according to standard procedures[13]. Subsequently, the eggs were analyzed by fluorescence microscopy (LSM800, Zeiss, Jena, Germany).

### Immunohistochemistry and signal distribution analyses

Testes were isolated from 1–2 days old males in PBS on concave microscope slides. They were transferred onto coverslips covered with Poly-L-lysine (Sigma-Aldrich, St. Louis, MO, USA) and positioned with an eyelash secured on a wooden stick. For immunostainings of isolated spermatids or sperm, the testes were further dissected with insect pins (Ø 0.15 mm) to uncover the respective cells. Specimens were fixed in 4% MeOH-free formaldehyde in PBS for 1 h, rinsed three times with PBS and permeabilized in 1% Triton X-100 in PBS for 1 h. With regard to anti-Nep4 antibody stainings, specimens were incubated with 0.15% SDS in PBS for 20 min

before blocking. Otherwise, specimens were blocked immediately after permeabilization in 1x ROTI ImmunoBlock (Carl Roth, Karlsruhe, Germany) for 40 min and incubated with primary antibodies at 4 °C overnight. Samples were washed with PBS three times for 10 min each and incubated with secondary antibodies at 4 °C overnight. In case of WGA and DAPI staining, dyes were applied with the secondary antibody solution. Finally, samples were washed as described before and mounted in Fluoromount-G mounting medium (SouthernBiotech, Birmingham, AL, USA). Primary antibodies were anti-Nep4 (RRID:AB_2569115, 1:200, monospecificity confirmed in ref. [13]), anti-GFP (RRID:AB_305564, 1:2000), anti-GFP (RRID:AB_300798, 1:500), anti-GM130 (RRID:AB_732675, 1:500), anti-GAPDH (TA337137, Origene, Rockville, MD, USA, 1:1000), and anti Golgin245 (RRID:AB_2569587, 1:200). The secondary antibodies were anti-rabbit-AF488 (RRID:AB_2338047, 1:200), anti-rabbit-AF633 (RRID:AB_2535733, 1:200), and anti-chicken-Cy3 (RRID:AB_2340363, 1:200). WGA-TRITC (Invitrogen, Carlsbad, CA, USA) and WGA-iFluor™647 (AAT Bioquest Inc., Pleasanton, CA, USA) were applied at a working concentration of 5 µg/ml. Nep4::mNG signal was preserved during the staining procedure, provided that no preincubation with SDS had taken place. Confocal images were captured with an LSM800 microscope (Zeiss) and further processed using Fiji ImageJ Software (National Institutes of Health, Bethesda, MD, USA). If not stated otherwise Z-stacks are displayed as maximum projections.

For signal distribution analyses, pixel intensities were quantified in maximum projection images using Fiji ImageJ Software. Fluorescent signal intensities for each channel were measured at a distance of 3.5 µm from the sperm tip towards the nucleus. WGA signals were used as a starting point. From this point, the mean pixel intensity of a width of 0.7 µm was calculated along the measured distance. At least 50 sperm isolated from the testes of at least five independent animals were measured for each genotype. The mean values ± SD were plotted against the distance from the sperm tip.

In the related statistical analyses, characteristic parameters of the individual signals were determined. For the Nep4::mNG, the Snky::GFP, and the WGA signals, the distance from the sperm tip to the respective maximum signal intensities was assessed. In addition, the distance from the sperm tip to the position at which the DAPI signal surpassed a predefined lower threshold value of 5 arbitrary units was evaluated. To measure the nucleus length, maximum projection images of the corresponding sperm were analyzed using Fiji ImageJ Software. For each genotype, a subset of at least 30 nuclei of sperm isolated from the testes of at least five individual animals was measured. The mean values ± SD were shown.

### Serial Block-Face Scanning Electron Microscopy (SBF-SEM)

Testes from 1-2 days old males were dissected in PBS and immediately fixed in 2.5% glutaraldehyde in 0.1 M cacodylate buffer (pH 7.4) for 1 h at RT, then stored at 4 °C for subsequent processing. Following an adapted version of the standard Ellisman protocol[33], samples were stained with heavy metals and infiltrated with resin.

In brief, after primary fixation, samples were post-fixed with 2% osmium tetroxide (Science Services, Munich, Germany) in 0.1 M cacodylate buffer (pH 7.4) for 1 h. Subsequently, they were transferred into 2.5% potassium ferrocyanide (Riedel de Haën, Seelze, Germany) in cacodylate for 30 min[34]. Both of these steps were performed on ice with gentle agitation. All further steps were carried out at RT with agitation and interspersed with washing steps in ddH$_2$O, repeated five times for 3 min each.

After rinsing, the samples were treated with thiocarbohydrazide (Riedel de Haën) in ddH$_2$O for 30 min, followed by an additional 2% osmication step in water for 1 h at RT. The samples were then stained with an aqueous 1% tannic acid (Riedel de Haën) solution for 20 min at RT. Subsequently, the samples were transferred into aqueous 1% uranyl acetate solution (Electron Microscopy Sciences, Hatfield, PA, USA) and kept at 4 °C overnight. On the following day, samples in uranyl acetate were heated to 50 °C for 30 min, rinsed with pre-warmed ddH$_2$O, then incubated in freshly prepared Walton's lead aspartate (Pb(NO$_3$)$_2$ (Carl Roth), L-Aspartate (Serva, Heidelberg, Germany), KOH (Merck, Darmstadt, Germany)) for 30 min at 60 °C.

The samples were then dehydrated through a graded ice-cold, anhydrous ethanol series (30%, 50%, 70%, and 90%) for 7 min each, followed by two 10-min rinses in pure ethanol. Further dehydration was carried out with chilled acetone followed by RT anhydrous acetone for 10 min each. Fully dehydrated samples were then infiltrated in an ascending Epon 812 (Sigma-Aldrich): acetone mixture (1:3, 1:1, 3:1) for 2 h, followed by an overnight incubation in pure hard resin mixture. The samples were then placed in a fresh Epon batch for another 6 h before being gently transferred onto absorbent paper to remove excess resin from the surface[35]. Finally, the samples were affixed to a sample rivet with two-component epoxy adhesive and polymerized for 48 h at 60 °C.

Once cured, rivets containing the samples were sputter-coated in 20 nm gold. Finally, samples were inserted into the 3View2XP Gatan stage (Gatan, Pleasanton, CA, USA), fitted in a JSM 7200 F (JEOL Ltd., Tokyo, Japan), and aligned parallel to the diamond knife-edge. Draining the resin from the surface of the testes earlier allowed precise navigation to the area of interest (Supplementary Fig. 8). Additionally, the samples' sensitivity to charging was drastically reduced, allowing higher accelerating voltages and longer dwell times without impairing the ability to cut at low slice thicknesses.

The samples remained stable under imaging conditions with an accelerating voltage of 3.1 kV, a 30 nm condenser aperture, high vacuum mode of 10 Pa, and a positive stage bias of 450 V. The imaging parameters were set to 4 nm pixel size, a 2.7 µs dwell time, 40 nm ablation interval, with the image size being typically at $8192 \times 10240$ pixels, according to the distribution and orientation of sperm cells. For each condition, approximate volumes of $25 \times 40 \times 12$ µm (comprising 300 slices) were acquired. Image acquisition was controlled through Gatan Digital Micrograph software (Version 3.32.2403.0). Subsequent postprocessing, including alignment, filtering, and segmentation, was carried out manually using Microscopy Image Browser (Version 2.9010[36],). The final visualization of the sperm tip, including the acrosome, was based on an interpolated slice across the complete image volume. Scaling of the images was based on the specific x-y slices corresponding to each individual cell. To optimize computational resource usage, the final stacks were binned before exporting the files for volumetric visualization in Amira 3D (Version 2022.1, Thermo Fisher, Waltham, MA, USA).

## On-section CLEM

Testes from 1-2 days old males were dissected in PBS and cryo-fixed using a HPF Compact 03 (Engineering Office M. Wohlwend GmbH, Sennwald, Switzerland). For this, testes were transferred into the 100 µm deep cavity of an aluminum planchette (no. 241), excess space was filled with 1-Hexadecen (Carl Roth, Karlsruhe, Germany), and the assembly was closed with another 100 µm deep planchette (no. 241) and immediately high-pressure frozen.

Subsequent freeze substitution and Lowicryl embedding was performed as described in ref. [37]. Samples were freeze substituted in 0.1% uranyl acetate (Science Services) in anhydrous acetone (VWR, Darmstadt, Germany) for 24 h at -90 °C. Then, the temperature was raised to −45 °C (5 °C/h) and samples were washed three times with anhydrous acetone. Next, infiltration with increasing concentrations (10%, 25%, 50%, 75%) of Lowicryl HM20 (Science Services) in acetone was carried out for 2 h each step. During the last two steps the temperature was raised by 10 °C each to −35 °C and −25 °C, respectively. Afterwards 100% Lowicryl was exchanged three times in 10 h steps. Finally, polymerization was carried out via UV light for 24 h at −25 °C and further 24 h at 20 °C. Polymerized specimen blocks were removed from the AFS, 250 nm thin sections were prepared using a Leica UC7 (Leica Microsystems, Wetzlar, Germany) and collected on 200 mesh carbon film grids (Plano, Wetzlar, Germany). For light microscopy, grids were placed on a 20 µl drop of PBS, pH 8 on a 25 mm coverslip. Grids were then sandwiched with another 25 mm coverslip and transferred into a custom-made holder. Z-stacks were recorded using an Olympus FV-3000 (Olympus, Hamburg, Germany) operated as a widefield setup equipped with a sCMOS camera (ORCA-Flash 4.0, Hamamatsu, Japan) and a 60x oil immersion lens (PLAPON-SC NA 1.4). Overview images were acquired

with a 10x objective (UPL SAPO NA 0.4) facilitating correlation later on within the TEM.

For tomogram acquisition, sections were labeled with 10 or 15 nm Protein-A gold fiducials on both sides. Afterwards, sections were contrasted with 3% uranyl acetate for 30 min and 2% lead citrate for 20 min in the LEICA EM AC20 (Leica, Wetzlar, Germany) and subsequently analyzed using a TEM 200 keV JEM2100-Plus (JEOL) equipped with a 20-megapixel EMSIS Xarosa CMOS camera (EMSIS, Münster, Germany). Regions of interest from light microscopy were relocated within the TEM and standard TEM images or tilt series from +-65° with 1° increments were acquired using the TEMography software (TEMography.com, JEOL). Tomograms were then reconstructed using the backprojection algorithm in IMOD[38]. Light microscopy images were finally overlaid on top of electron microscopy data using the Shot Meister software (TEMography.com, JEOL).

## Pull-down analyses and Western blot

20 one-week-old male flies of the control genotype (*bam*-Gal4 > UAS-roGFP), or the sample genotype (*bam*-Gal4 > UAS-Nep4::roGFP) were frozen in liquid nitrogen and homogenized. Homogenates were resuspended in 300 µl lysis buffer (150 mM NaCl, 50 mM Tris, 1 mM MgCl, 1% n-dodecyl-β-D-maltopyranoside [DDM]) and centrifuged ($10000 \times g$, 15 min). The resulting supernatants were incubated for 30 min at 4 °C with 50 µl of µMACS anti-GFP MicroBeads (Miltenyi Biotec, Auburn, CA, USA). Beads were immobilized in calibrated µ-columns (Miltenyi Biotec), equilibrated beforehand with 200 µl lysis buffer. After two washing steps using 200 µl lysis buffer, bound proteins were reduced by incubation in 50 µl reducing solution (10 mM dithiothreitol [DTT], 100 mM NH$_4$HCO$_3$) for 5 min at RT and an additional 30 min at 38 °C, followed by two washing steps with 20 mM Tris-HCl (pH 7.5). Subsequently, bound proteins were alkylated by adding 50 µl alkylation solution (54 mM Iodacetamid, 100 mM NH$_4$HCO$_3$) and incubation for 15 min in the dark. The columns were then washed twice with 100 µl digestion buffer (50 mM NH$_4$HCO$_3$, 5% acetonitrile), sealed, and incubated with 25 µl trypsin solution (0.01 µg trypsin / lysC in digestion buffer) overnight. Eluates were centrifuged ($10000 \times g$, 10 min), and the supernatant was subjected to mass spectrometry analyses (5 µl per sample, see below). At least three independent biological replicates were analyzed per genotype.

For Western blot analyses, 40 testes per replicate and genotype were homogenized (glass-teflon homogenizer) in PBS containing protease inhibitor mix M (Serva, Heidelberg, Germany). Subsequently, samples were boiled at 99 °C for 3 min in Laemmli buffer. Protein samples (10 µg/lane) were separated by SDS-PAGE (12%) and transferred to nitrocellulose membranes. Immunodetection was done using anti-Nep4 (1:2000) and anti-GAPDH antibodies (1:1000), respectively.

## Enzymatic cleavage assays

Heterologous expression of His-tagged (10x) or mNeonGreenHis-tagged Nep4B (soluble isoform) was performed in *Sf*21 cells as previously described[16]. Briefly, transfected (TransIT®-Insect Transfection Reagent, Mirus Bio, Madison, WI, USA) and non-transfected *Sf*21 cells were cultured in 75 cm$^2$ flasks for 72 h and harvested by centrifugation ($300 \times g$, 5 min). Subsequently, cells were resuspended in 5 ml binding buffer (50 mM NaH$_2$PO$_4$, pH 7.9, 300 mM NaCl) and lysed with a glass-teflon homogenizer. The resulting homogenates were centrifuged (10000 x g, 10 min), and the supernatants were subjected to gravity-flow-based His-tag purification (Protino Ni-NTA agarose, Macherey-Nagel, Düren, Germany) according to the manufacturer's instructions.

To measure enzymatic activity, 3 µl of Nep4-containing (10 ng/ml, purified from *nep4-His* or *nep4-mNeonGreenHis* transfected cells) and non-containing (from untransfected control cells) preparations were supplemented with 7 µl (250 ng) of SCLA peptide. All dilutions were prepared in hydrolysis buffer (100 mM NaCl, 50 mM Tris, pH 7.0). After 5 h of incubation (35 °C), 1 µl of each respective preparation was analyzed via mass spectrometry (see below). The SCLA peptide (luminal part, YLIYAVLa) was

synthesized at JPT Peptide Technologies (Berlin, Germany) with more than 90% purity. Individual cleavage assays were repeated at least three times.

## Mass spectrometry and data analysis

Dried peptides were resuspended in 10 μl LC-Load (PreOmics, Planegg, Germany), resulting in a protein concentration of 100 ng/μl. 1ul was used to perform reversed-phase chromatography on a Thermo Ultimate 3000 RSLC nano system connected to a TimsTOF HT mass spectrometer (Bruker Corporation, Bremen, Germany) through a Captive Spray ion source. Peptides were separated on an Aurora Gen3 C18 column (25 cm x 75um x 1.6um) with CSI emitter (Ionopticks, Collingwood, Australia) at a temperature of 40 °C. Elution of peptides from the column was realized via a linear gradient of acetonitrile from 2-35% (in water with 0.1% formic acid) for 44 min at a constant flow rate of 300 nl/min following a 7 min increase to 50%, and finally, 4 min to reach 85%. Eluted peptides were directly electro sprayed into the mass spectrometer at an electrospray voltage of 1.5 kV and 3 l/min dry gas. The MS settings of the timsTOF were adjusted to positive ion polarity with a MS range from 100 to 1700 m/z. The scan mode was set to DDA-PASEF. The ion mobility was ramped from 0.7 Vs/cm$^2$ to 1.5 in 100 ms. The accumulation time was also set to 100 ms. 10 PASEF ramps per cycle resulted in a duty cycle time of 1.17 s. The target intensity was adjusted to 14000, the intensity threshold to 1200. The dynamic exclusion time was set to 0.4 min to avoid repeated scanning of the precursor ions, their charge state was limited from 0 to 5.

The data were loaded to PeaksOnline Version 12 (Bioinformatics Solutions Inc., Waterloo, Ontario, Canada) and analysed with the PeaksOnline workflow (DeNovo and DB Search) with precursor mass error tolerance of 15 ppm, fragment mass error tolerance of 0.5 Da, CSS error tolerance of 0.05, and a missed cleavage of 2. As modification carbamidomethylation (C) (fix) and oxidation (M) (variable) were chosen. As protein database the *Drosophila*-specific UP000000803 was used (www.uniprot.org/proteomes/UP000000803). The results were filtered for peptides with an FDR of 1%, for proteins by the -10lgP of 20 and significance was calculated using PEAKS software (one-way ANOVA). Label-free quantification was conducted by comparing peptide and protein amounts of different groups according to established protocols[39], with each group consisting of a minimum of three independent biological replicates.

## Statistics and reproducibility

Unless otherwise indicated, one-way ANOVA followed by Tukey's Multiple Comparison Test was employed for statistical analyses. For the test, a $p$ value < 0.05 was considered significant (*$p < 0.05$, **$p < 0.01$, ***$p < 0.001$). All data were analyzed and visualized using GraphPad Prism (Version 10.3.0, GraphPad Software Inc., San Diego, CA, USA). For all boxplots, the center line of a plot indicates the median; the upper and lower bounds indicate the 75th and 25th percentiles, respectively; and the whiskers indicate the minimum and maximum. The proteomics data were statistically analyzed and visualized with PEAKS Studio software. All experiments were performed at least in triplicates. Western blot quantifications were statistically analyzed using a paired t-test (two-tailed).

## Data availability

The data supporting the findings from this study are available within the manuscript and its supplementary information. Source data are provided with this paper (Supplementary Data 1, Supplementary Data 2).

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

## Acknowledgements

We thank Martina Biedermann, Kerstin Etzold, Mechthild Krabusch, and Birgit Hemmis for excellent technical assistance. We further acknowledge the integrated Bioimaging facility (iBiOs) within the Center for Cellular Nanoanalytics (CellNanOs) at Osnabrück University for electron microscopy support, Marvin Kruse for performing pull-down analyses, and the Vienna *Drosophila* Resource Center (VDRC) and the Bloomington *Drosophila* Stock Center (BDSC, NIH P40OD018537) for providing fly stocks. A fly line expressing GFP-tagged Snky protein was a kind gift from Dr. Barbara T. Wakimoto (University of Washington, USA). This work was supported by grants from the Deutsche Forschungsgemeinschaft to H.M. (HA 6421/4-1; SFB 944, TP21), to A.P. (PA 517/12-1, 12-2; SFB 944, TP7 and SFB 1557, TP12), to O.E.P. (SFB 1557, Z2), by a stipend from Osnabrück University (A.B.), and by the Open Access Publishing Fund of Osnabrück University.

## Author contributions

A.B.: conception and design, acquisition of data, analysis and interpretation of data, drafting or revising the article; M.S.: acquisition of data, analysis and interpretation of data, drafting or revising the article; E.C.: acquisition of data, analysis and interpretation of data, drafting or revising the article; L.B.: acquisition of data, analysis and interpretation of data, drafting or revising the article; R.F.: acquisition of data, analysis and interpretation of data, drafting or revising the article; I.K.: acquisition of data, analysis and interpretation of data, drafting or revising the article; O.E.P.: acquisition of data, analysis and interpretation of data, drafting or revising the article; S.W.: acquisition of data, analysis and interpretation of data, drafting or revising the article; A.P.: acquisition of data, analysis and interpretation of data, drafting or revising the article; H.M.: conception and design, acquisition of data, analysis and interpretation of data, drafting or revising the article.

## Funding

## Competing interests

The authors declare no competing interests.
