## [Transparent Peer Review file · Communications Biology]

Nepriylsin 4 controls acrosome structure and male fertility in *Drosophila melanogaster*

Corresponding Author: Dr Heiko Meyer

Version 0:

Reviewer comments:

Reviewer #1

(Remarks to the Author)

The manuscript by Buhr et al. expands on earlier reports suggesting that the *Drosophila* endopeptidase Nepriylsin 4 (Nep4) is essential for male spermatogenesis, as Nep4-deficient male flies are sterile. In the present study, the authors propose that Nep4 localizes to the acrosome and is involved in establishing correct acrosome morphology, sperm tip structure, and male fertility.

To support their conclusions, the authors provide the following evidence:

- (i) Homozygous male flies expressing a mNeonGreen (mNG)-tagged Nep4 fusion protein with impaired catalytic activity are sterile, similar to Nep4-deficient males;
- (ii) Nep4 localizes mainly to the apical tip of individualized sperm, corresponding to the site of emerging acrosome;
- (iii) The acrosome of homozygous mNG::Nep4 males exhibits impaired morphology, as evidenced by a collapsed acrosomal area with reduced suborganellar patterning, as well as reduced length and volume.

The manuscript is of a high quality, it is generally well-written, and the figures are prepared with great attention to detail.

However, although the study is a carefully conducted and well-documented evidence supports the authors' interesting hypothesis, there are a few conceptual and technical issues that need to be addressed to improve its clarity.

Major points:

1. One of the control used in the manuscript is nep4-specific knockdown line. Since these flies have been used in previous studies, the authors opted not to confirm Nep4 expression reduction in this study. They cite three references, including Panz et al. (2012)., to support the efficacy of this knockdown line. However, from my understanding Panz et al. performed their knockdown experiment at 29°C, which is a standard temperature for inducing RNAi in flies, whereas the authors conducted their experiments at 27°C. I understand that this choice can stem from the fact that male fertility declines at higher temperatures, which could directly affect the study. However, the authors should provide the evidence that the level of knockdown at 27°C is sufficient.
2. The level of Nep4 expression appears to be significantly lower in Nep4::mNG homozygous flies compared to w1118 (Supplemental Figure 1). Could the authors provide quantitative data to confirm this observation? How does it compare to the RNAi line?
3. Although RNAi flies exhibit impaired fertility (Fig.1), their fertility is still higher than that of Nep4::mNGhomo flies. Is it due to residual Nep4 expression in RNAi flies? The authors should include statistical analysis comparing the fertility of these two lines and discuss this in the context of Nep4 expression levels in the RNAi line. This would address my previous questions about the knockdown efficacy.

Minor points:

1. It is interesting that C-terminal fusion protein leads to such a strong infertility phenotype and can be successfully used to investigate the role of Nep4 in *Drosophila* spermiogenesis without relying on knockout/knockdown male flies. Since the enzymatic activity of Nep4 fused with mNG is impaired, could the authors provide more details in the manuscript on why it occurs? How does the C-terminal fusion affect the enzyme function, considering the location of catalytic domains and overall structure? If the infertility phenotype is specific only to the C-terminal fusion, and as I understand, not to N-terminal fusion, it would be interesting to see authors' ideas on this. Specifically, if the differences result solely from the loss of enzymatic activity rather than due to altered expression levels (Major points 1-3) or crucial protein-protein interactions.
2. The videos attached to the manuscript are not clearly labeled – the information on the specific genotype shown and analyzed is missing.
3. The discussion is too lengthy and repetitive and should be shortened by at least half. There is no need to refer to figures again in this section. Additionally, lines 420-430 should be placed in the Results section rather than the Discussion.

4. Fig. 2 A' is mislabeled – if the control flies were w1118, they do not express mNG. Moreover, the figure legend says “No mNG signal (mNG) above background is observed in control animals”. This is somewhat confusing – do the authors mean there is no background fluorescence in the green channel? Since w1118 flies do not express mNG, this should be clarified.
5. The sentence in the introduction (lines 60-61) “It emerges from the acroblast during spermiogenesis and originates from the Golgi apparatus” is a bit awkward. It is the acroblast, which originates from the Golgi, that gives rise to the acrosome.
6. The last paragraph of the introduction (lines 77-95) is too long and should be significantly shortened.
7. Citations in the text should appear after the first sentence in which they are relevant, rather than at the end of the whole paragraph. For example, the citation 14 should appear in line 73 rather than line 76, as this improves clarity, especially when following references while reading. Could the authors ensure that this is applied consistently throughout the manuscript?

Reviewer #2

(Remarks to the Author)

In this manuscript, Buhr et al. present Neprilysin 4 as a potential protein that plays a crucial role in organizing the acrosome's structure and enhancing male fertility. The researchers demonstrated that secreted Neprilysin 4 localizes to the acrosome through immunostaining and by tagging the gene with Neon-green. However, the tagging of Neprilysin 4 interferes with the protein's function, leading to male sterility. To characterize the function of the neprilysin 4 gene, the authors employed RNA interference (RNAi) to downregulate the gene product, which revealed structural abnormalities in the acrosomes. Nonetheless, several technical issues within the study raise doubts about the validity of the conclusions drawn.

My main critical points are the following:

1. The effectiveness of RNA interference (RNAi) in the germline has not been clearly established. The efficiency of the RNAi line employed was previously assessed solely in somatic tissues, with no testing conducted in the testis as indicated in the referenced papers. KK RNAi lines may be effective in the germ line by using UAS-Dicer2 overexpression.
2. The reason for the observed fertility of w1118-bam-Gal4 being twice that of w1118 alone remains unclear. There is a lack of data regarding the effectiveness of Nep4 downregulation (bam>nep4RNAi) through the use of anti-Nep4 antibodies or qRT-PCR. This is crucial for the characterization of the function of the nep4 gene.
3. CRISPR technology enables the deletion of the gene segment that encodes the extracellular portion of the protein. However, it is uncertain whether tagging the protein results in a neomorphic effect or a gain-of-function mutation. A definitive deletion in the same gene region would provide stronger evidence.
4. It is not addressed that male sterility is due to abnormal spermiogenesis or abnormal fertilization. Do the seminal vesicles or the spermatheca of the females used in the fertility test show Nep4-mNG signal in the homozygotes Nep4-mNG?
5. The colocalization of the antibody with Nep4-mNG has not been demonstrated in the same spermatids or sperm samples. What is the localization of the protein as determined by antibody staining during the earlier phases of spermatogenesis?
6. It is not clear how many independent testes (animals) were analyzed in each experiment (50 individual sperm/spermatids could be in a single seminal vesicle/testis in a single animal Fig.2.,3.,4.). Acroblast formation typically begins in round spermatids; however, the localization pattern of Nep4-mNG does not coincide with that of GM130 in any stages of spermatogenesis.
7. The differences in the localization pattern of Nep4-mNG and acrosome morphology between heterozygotes and homozygotes presented in Fig. 3 are not compelling.
8. If the antibody against Nep4 is effective in the testis, it raises the question of why it has not been utilized for colocalization studies with snky-GFP.
9. In the analysis of sperm tips, it is necessary to measure and present the entire sperm nuclei, not just part of it (Fig. 5).
10. The evaluation of acrosome structural modifications illustrated in Figure 6 could be deceptive, given the considerable differences in nucleus lengths across the various genotypes presented in Fig. 6A. It is unclear whether the measurements correspond to mature sperm or elongated spermatids, indicating that the variations in acrosomal size might result from differing developmental stages. Furthermore, which genotype is utilized as the control in this experiment?
11. Globozoospermia is a human disorder, and the conclusion that Nep4-mNG is akin to globozoospermia is an overreaching statement.

Reviewer #3

(Remarks to the Author)

The manuscript from Buhr et al describes a study aimed at elucidating the role(s) of the peptidase Neprilysin 4 (Nep4) during spermatogenesis in *Drosophila*. Previous work from the authors and other groups had established that Nep4 is expressed during spermatogenesis and that Nep4 function is critical for male fertility. While Nep4-deficient sperm appear to develop normally and are capable of fertilizing eggs, they fail to promote any measure of embryonic development. However, detailed and in-depth analysis of Nep4 function during spermatogenesis has been lacking.

In order to proceed, the authors generated Nep4::mNG, an endogenously tagged variant of Nep4, in which the fluorescent marker mNeonGreen is attached to the C-terminus of the protein. This construct enabled them to both follow the localization of Nep4 during *Drosophila* spermatogenesis and to examine the mutant phenotype, as Nep4::mNG appears to be non-functional. The major findings made were as follows:

1. Nep4 exhibits a highly specific localization to the developing and mature sperm acrosome.
2. The acrosome is divided into several distinct domains, with Nep4::mNG localizing anteriorly to a second acrosomal marker, Snky-GFP.
3. Disruption of Nep4 function results in acrosomal abnormalities: the normally elongated acrosome takes on a more

rounded morphology, and the partitioning of the acrosome and sperm tip into distinct domains are lost.
4. The severe phenotypes observed in *Nep4::mNG* homozygous males, which encodes a stable but non-functional peptidase, implies that the mutant phenotypes are indeed the result of absence of peptidase capacity.

This is a very well-executed study. The authors make highly competent use of different and complementary light and electron-microscopy techniques, along with careful quantitation of their data. The assignment of an acrosomal context for *Nep4* function is novel and significant, and helps define clear avenues for future research. That said, the overall advance made in ascribing a defined function for *Nep4* during spermatogenesis is rather minimal and does not go beyond focusing attention on the acrosome, without providing mechanistic insight into the manner by which *Nep4* peptidase activity contributes to proper acrosome formation and organization, and why the observed acrosomal abnormalities cause such a strong disruption of the capacity of *Nep4*-deficient sperm to fertilize eggs and initiate embryogenesis. Sneaky mutant sperm are capable of penetrating eggs, indicating that the acrosome in *Drosophila* may not be essential for sperm penetration. However, the sneaky mutant sperm nucleus fails to undergo decondensation after egg penetration. Further characterization of *Nep4*-deficient sperm following egg penetration would significantly enhance the manuscript.

Version 1:

Reviewer comments:

Reviewer #1

(Remarks to the Author)

The authors have satisfactorily addressed all of my comments and concerns. I appreciate the effort they put in revising the manuscript and performing additional experiments to strengthen their conclusions. Therefore, I recommend the manuscript for publication in its current form.

Reviewer #2

(Remarks to the Author)

Reviewer #2 (Remarks to the Author):

The originality of the current paper is constrained, as Ohsako, T. et al. have previously detailed the male sterile phenotype. The primary novelty of this paper lies in the localization of *Nep4* to the acrosome; however, the absence of colocalization or co-staining with other acrosomal proteins or organelles and along with the inadequately presented new experimental data, undermines the robustness of the present research.

1. The effectiveness of RNA interference (RNAi) in the germline has not been clearly established. The efficiency of the RNAi line employed was previously assessed solely in somatic tissues, with no testing conducted in the testis as indicated in the referenced papers. KK RNAi lines may be effective in the germ line by using UAS-Dicer2 overexpression.

We assessed and confirmed a high testis-specific knockdown efficiency by western blot. Corresponding data are shown in a novel Supplementary Fig. 3.

The experiment demonstrating the effectiveness of RNA interference is questionable; it certainly does not show strong silencing efficiency. The objectivity of the evaluation is problematic. Ponceau S staining cannot be used for protein quantification in this form. In addition to presenting the entire Ponceau-stained membrane, not just a small part of it (which is barely visible), it is essential to detect a ubiquitously expressed protein with a specific antibody (loading control). The size of the protein allows the membrane to be cut, and the amount of protein applied could be verified with a well-established anti-actin or anti-tubulin antibody. Details of this experiment are missing from the materials and methods section. The molecular weight marker is barely visible.

2. The reason for the observed fertility of *w1118-bam-Gal4* being twice that of *w1118* alone remains unclear.

In contrast to the *w1118* and *Nep4::mNG* animals, which were reared at 22 °C (RT), all knockdown experiments (including the controls) were performed at 27 °C. This difference in temperature likely accounts for the increased number of adult offspring. Corresponding information are included in the Materials and Methods section.

It is not entirely clear what is meant by "including the controls."

The *w1118* could be a control, similarly to the *Nep4*-RNAi line alone or the *bam-Gal4* line at 27 °C.

What is the scientific evidence or explanation for the increased number of adult offspring at higher temperatures?

Why wasn't 27°C used for all experiments regardless of the genotype?

There is a lack of data regarding the effectiveness of *Nep4* downregulation (*bam>nep4RNAi*) through the use of anti-*Nep4* antibodies or qRT-PCR. This is crucial for the characterization of the function of the *nep4* gene.

We assessed and confirmed a high testis-specific knockdown efficiency by western blot. Corresponding data are shown in a novel Supplementary Fig. 3.

My answer to the first question applies here as well.

3. CRISPR technology enables the deletion of the gene segment that encodes the extracellular portion of the protein. However, it is uncertain whether tagging the protein results in a neomorphic effect or a gain-of-function mutation. A definitive deletion in the same gene region would provide stronger evidence.

The fact that the RNAi knockdown experiments phenocopied the effects of the tagging strongly indicates a loss-of-function effect. Corresponding information were included into the results section.

As established by Ohsako, T. et al., it has been demonstrated that Nep4 is crucial for normal fertilization; thus, the originality of the current paper must be substantiated through experimental evidence. A mutant with a deletion at the C-terminal could validate the essential role of Nep4 in maintaining normal fertility. Additionally, an N-terminally tagged variant of Nep4, created using CRISPR technology, would further corroborate these findings. The materials and methods section fails to specify which gRNA was utilized; could this potentially lead to off-target effects in addition to tagging Nep4? The same gRNA might also be employed to produce a loss-of-function mutant. It is not addressed that male sterility is due to abnormal spermiogenesis or abnormal fertilization. Do the seminal vesicles or the spermatheca of the females used in the fertility test show Nep4-mNG signal in the homozygotes Nep4-mNG?

Seminal vesicles showing Nep4-mNG signals are depicted in Fig. 2B; presence of individualized sperm in the seminal vesicles indicates proper completion of spermatogenesis. The text was amended accordingly. Further analyses on the seminal receptacles revealed that Nep4::mNG sperm from homozygous males are present in the female storage organs, however, they are discarded rather quickly (relative to the heterozygous control). Moreover, we found that the proportion of fertilized oocytes was significantly reduced under homozygous conditions, relative to the heterozygous control, and that none of the laid eggs initiated embryonic development, even if sperm entry had occurred. The resulting data are presented in a new Figure 7.

Answer accepted.

5. The colocalization of the antibody with Nep4-mNG has not been demonstrated in the same spermatids or sperm samples. What is the localization of the protein as determined by antibody staining during the earlier phases of spermatogenesis?

Since the Nep4 antibody requires a preincubation with SDS to produce a significant signal in tissue stainings, we were not able to do Nep4 / mNG co-stainings in the same samples (SDS incubation was incompatible with the mNG signal detection). Corresponding information are provided in the Materials and Methods section.

Given the high similarity of the Nep4 and mNG signal patterns (Figure S5), we concluded that Nep4::mNG exhibits a subcellular localization identical to that of the untagged protein. Thus, we rather analyzed the mNG signal in detail. No signal was detected in phases prior to early elongating spermatids.

Answer accepted.

6. It is not clear how many independent testes (animals) were analyzed in each experiment (50 individual sperm/spermatids could be in a single seminal vesicle/testis in a single animal Fig.2.,3.,4.).

All experiments were based on testes from at least five independent animals. Corresponding information were included into the respective figure legends.

The use of five independent animals in an experiment is very minimal, but a statement of the independent experiments in the materials and methods section would also be important for assessing reproducibility.

Acroblast formation typically begins in round spermatids; however, the localization pattern of Nep4-mNG does not coincide with that of GM130 in any stages of spermatogenesis.

We agree that the two signals do not overlap in early elongating spermatids. However, up to now we have not analyzed any additional stages of spermatogenesis regarding GM130 / Nep4-mNG co-distribution.

It is interesting and important to find out the strong localization of Nep4 in the early elongating spermatids. Lysosomal, trans-Golgi, or basal body staining can help to specify better the nature of the signal.

7. The differences in the localization pattern of Nep4-mNG and acrosome morphology between heterozygotes and homozygotes presented in Fig. 3 are not compelling.

We removed the corresponding statements for the CLEM analyses (Fig. 3A) and attenuated them for the fluorescence stainings (Fig. 3B). Quantification of the relevant morphological defects was done in Figs. 4 and 6.

Answer accepted.

8. If the antibody against Nep4 is effective in the testis, it raises the question of why it has not been utilized for colocalization studies with snky-GFP.

The Nep4 antibody requires a preincubation with SDS to produce a significant signal in tissue stainings; this procedure was not compatible with the GFP signal detection (see above). Corresponding information are provided in the Materials and

Methods section.
Answer accepted.

9. In the analysis of sperm tips, it is necessary to measure and present the entire sperm nucleus, not just part of it (Fig. 5).

This question is not answered.

As is stated, "50 individual sperm cells isolated from the testes". Using sperm from the seminal vesicles in this experiment would be more objective, including the entire sperm nucleus in the measurement, not just part of it.

10. The evaluation of acrosome structural modifications illustrated in Figure 6 could be deceptive, given the considerable differences in nucleus lengths across the various genotypes presented in Fig. 6A. It is unclear whether the measurements correspond to mature sperm or elongated spermatids, indicating that the variations in acrosomal size might result from differing developmental stages. Furthermore, which genotype is utilized as the control in this experiment?

To ensure analysis of similar developmental stages, we analyzed sperm at exactly the same position within the respective testes (see Supplementary Fig. 8). Furthermore, we selected sperm with nuclei devoid of any organelles surrounding them, indicating a largely completed individualization. For the light microscopic analyses (Figs. 4, 5, S7) only individualized sperm that were not surrounded by cyst cells were selected.

w1118 x UAS-nep4RNAi was used as a control. Corresponding information were included into the figure legend.

Spermatogenesis is such a dynamic process; therefore, even in 24 hours, stages and distribution of cysts are changing, therefore, "exactly the same position" does not necessarily mean the same stages.

11. Globozoospermia is a human disorder, and the conclusion that Nep4-mNG is akin to globozoospermia is an overreaching statement.

The statement has been removed from the abstract and attenuated in the discussion section.

Answer accepted.

References

Ohsako, T. et al. The *Drosophila* Nepriysin 4 gene is essential for sperm function following sperm transfer to females. *Genes Genet. Syst.* 96, 21–00024 (2021).

Reviewer #3

(Remarks to the Author)

Following the revisions and additions made, I find that the manuscript has significantly improved and is now suitable for publication in your journal

Version 2:

Reviewer comments:

Reviewer #2

(Remarks to the Author)

Following the modifications, I conclude that the manuscript has markedly improved and is now ready for publication.

Reviewer #1 (Remarks to the Author):

The manuscript by Buhr et al. expands on earlier reports suggesting that the *Drosophila* endopeptidase Neprilysin 4 (Nep4) is essential for male spermatogenesis, as Nep4-deficient male flies are sterile. In the present study, the authors propose that Nep4 localizes to the acrosome and is involved in establishing correct acrosome morphology, sperm tip structure, and male fertility. To support their conclusions, the authors provide the following evidence:

- (i) Homozygous male flies expressing a mNeonGreen (mNG)-tagged Nep4 fusion protein with impaired catalytic activity are sterile, similar to Nep4-deficient males;
- (ii) Nep4 localizes mainly to the apical tip of individualized sperm, corresponding to the site of emerging acrosome;
- (iii) The acrosome of homozygous mNG::Nep4 males exhibits impaired morphology, as evidenced by a collapsed acrosomal area with reduced suborganellar patterning, as well as reduced length and volume.

The manuscript is of a high quality, it is generally well-written, and the figures are prepared with great attention to detail. However, although the study is a carefully conducted and well-documented evidence supports the authors' interesting hypothesis, there are a few conceptual and technical issues that need to be addressed to improve its clarity.

Major points:

1. One of the controls used in the manuscript is nep4-specific knockdown line. Since these flies have been used in previous studies, the authors opted not to confirm Nep4 expression reduction in this study. They cite three references, including Panz et al. (2012), to support the efficacy of this knockdown line. However, from my understanding Panz et al. performed their knockdown experiment at 29°C, which is a standard temperature for inducing RNAi in flies, whereas the authors conducted their experiments at 27°C. I understand that this choice can stem from the fact that male fertility declines at higher temperatures, which could directly affect the study. However, the authors should provide the evidence that the level of knockdown at 27°C is sufficient.

We assessed and confirmed a high testis-specific knockdown efficiency at 27 °C by western blot. Corresponding data are shown in a novel Supplementary Fig. 3.

2. The level of Nep4 expression appears to be significantly lower in Nep4::mNG homozygous flies compared to w¹¹¹⁸ (Supplemental Figure 1). Could the authors provide quantitative data to confirm this observation? How does it compare to the RNAi line?

We analyzed the individual Nep4 protein levels by western blot and did not observe any significant differences between homozygous Nep4::mNG flies and w¹¹¹⁸ control flies. A new representative blot as well as the respective quantifications of three biological replicates are depicted in an amended Supplementary Fig. 1.

3. Although RNAi flies exhibit impaired fertility (Fig.1), their fertility is still higher than that of Nep4::mNGhomo flies. Is it due to residual Nep4 expression in RNAi flies? The authors should include statistical analysis comparing the fertility of these two lines and discuss this in the context of Nep4 expression levels in the RNAi line. This would address my previous questions about the knockdown efficacy.

Our analyses on the knockdown efficiency indeed revealed a residual *nep4* expression in the RNAi flies (now shown in a new Supplementary Fig. 3), which presumably explains the differences in fertility between the RNAi and the Nep4::mNG flies. However, since the w¹¹¹⁸ and Nep4::mNG animals were reared at 22 °C (RT), while the knockdown experiments (including the controls) were performed at 27 °C (due to the temperature sensitivity of the UAS-Gal4 system), we consider it problematic to directly compare these conditions.

Information on the individual temperatures are included in the Materials and Methods section.

Minor points:

1. It is interesting that C-terminal fusion protein leads to such a strong infertility phenotype and can be successfully used to investigate the role of Nep4 in *Drosophila* spermiogenesis without relying on knockout/knockdown male flies. Since the enzymatic activity of Nep4 fused with mNG is impaired, could the authors provide more details in the manuscript on why it occurs? How does the C-terminal fusion affect the enzyme function, considering the location of catalytic domains and overall structure? If the infertility phenotype is specific only to the C-terminal fusion, and as I understand, not to N-terminal fusion, it would be interesting to see authors' ideas on this. Specifically, if the differences result solely from the loss of enzymatic activity rather than due to altered expression levels (Major points 1-3) or crucial protein-protein interactions.

Among the different functional motifs of neprilysins, the "CxxW" motif is characteristically located at the very C-terminus. Since this motif has been reported to be essential to protein folding and maturation, fusion of a rather large tag at that position likely impairs proper function of the enzymes. Corresponding information were included into the Results section.

2. The videos attached to the manuscript are not clearly labeled – the information on the specific genotype shown and analyzed is missing.

The genotypes were embedded into the corresponding videos.

3. The discussion is too lengthy and repetitive and should be shortened by at least half. There is no need to refer to figures again in this section. Additionally, lines 420-430 should be placed in the Results section rather than the Discussion.

The discussion was considerably shortened.

Given the limited amount of experimental data that could be deduced from the pull-down experiment, we prefer to present and discuss the data here (lines 420-430) rather than in the Results section.

4. Fig. 2 A' is mislabeled – if the control flies were w1118, they do not express mNG. Moreover, the figure legend says "No mNG signal (mNG) above background is observed in control animals". This is somewhat confusing – do the authors mean there is no background fluorescence in the green channel? Since w1118 flies do not express mNG, this should be clarified.

The incorrect labeling was corrected and the figure legend was clarified. Indeed, we wanted to emphasize that there was no background fluorescence in the green channel.

5. The sentence in the introduction (lines 60-61) "It emerges from the acroblast during spermiogenesis and originates from the Golgi apparatus" is a bit awkward. It is the acroblast, which originates from the Golgi, that gives rise to the acrosome.

The sentence has been rephrased.

6. The last paragraph of the introduction (lines 77-95) is too long and should be significantly shortened.

The paragraph was amended accordingly.

7. Citations in the text should appear after the first sentence in which they are relevant, rather than at the end of the whole paragraph. For example, the citation 14 should appear in line 73 rather than

line 76, as this improves clarity, especially when following references while reading. Could the authors ensure that this is applied consistently throughout the manuscript?

The manuscript was amended accordingly.

Reviewer #2 (Remarks to the Author):

In this manuscript, Buhr et al. present Neprilysin 4 as a potential protein that plays a crucial role in organizing the acrosome's structure and enhancing male fertility. The researchers demonstrated that secreted Neprilysin 4 localizes to the acrosome through immunostaining and by tagging the gene with Neon-green. However, the tagging of Neprilysin 4 interferes with the protein's function, leading to male sterility. To characterize the function of the neprilysin 4 gene, the authors employed RNA interference (RNAi) to downregulate the gene product, which revealed structural abnormalities in the acrosomes. Nonetheless, several technical issues within the study raise doubts about the validity of the conclusions drawn.

My main critical points are the following:

1. The effectiveness of RNA interference (RNAi) in the germline has not been clearly established. The efficiency of the RNAi line employed was previously assessed solely in somatic tissues, with no testing conducted in the testis as indicated in the referenced papers. KK RNAi lines may be effective in the germ line by using UAS-Dicer2 overexpression.

We assessed and confirmed a high testis-specific knockdown efficiency by western blot. Corresponding data are shown in a novel Supplementary Fig. 3.

2. The reason for the observed fertility of w1118-bam-Gal4 being twice that of w1118 alone remains unclear.

In contrast to the w¹¹¹⁸ and Nep4::mNG animals, which were reared at 22 °C (RT), all knockdown experiments (including the controls) were performed at 27 °C. This difference in temperature likely accounts for the increased number of adult offspring. Corresponding information are included in the Materials and Methods section.

There is a lack of data regarding the effectiveness of Nep4 downregulation (bam>nep4RNAi) through the use of anti-Nep4 antibodies or qRT-PCR. This is crucial for the characterization of the function of the nep4 gene.

We assessed and confirmed a high testis-specific knockdown efficiency by western blot. Corresponding data are shown in a novel Supplementary Fig. 3.

3. CRISPR technology enables the deletion of the gene segment that encodes the extracellular portion of the protein. However, it is uncertain whether tagging the protein results in a neomorphic effect or a gain-of-function mutation. A definitive deletion in the same gene region would provide stronger evidence.

The fact that the RNAi knockdown experiments phenocopied the effects of the tagging strongly indicates a loss-of-function effect. Corresponding information were included into the results section.

4. It is not addressed that male sterility is due to abnormal spermiogenesis or abnormal fertilization. Do the seminal vesicles or the spermatheca of the females used in the fertility test show Nep4-mNG signal in the homozygotes Nep4-mNG?

Seminal vesicles showing Nep4-mNG signals are depicted in Fig. 2B; presence of individualized sperm in the seminal vesicles indicates proper completion of spermatogenesis. The text was amended accordingly.

Further analyses on the seminal receptacles revealed that Nep4::mNG sperm from homozygous males are present in the female storage organs, however, they are discarded rather quickly (relative to the heterozygous control). Moreover, we found that the proportion of fertilized oocytes was significantly reduced under homozygous conditions, relative to the heterozygous control, and that none of the laid eggs initiated embryonic development, even if sperm entry had occurred. The resulting data are presented in a new Figure 7.

5. The colocalization of the antibody with Nep4-mNG has not been demonstrated in the same spermatids or sperm samples. What is the localization of the protein as determined by antibody staining during the earlier phases of spermatogenesis?

Since the Nep4 antibody requires a preincubation with SDS to produce a significant signal in tissue stainings, we were not able to do Nep4 / mNG co-stainings in the same samples (SDS incubation was incompatible with the mNG signal detection). Corresponding information are provided in the Materials and Methods section.

Given the high similarity of the Nep4 and mNG signal patterns (Figure S5), we concluded that Nep4::mNG exhibits a subcellular localization identical to that of the untagged protein. Thus, we rather analyzed the mNG signal in detail. No signal was detected in phases prior to early elongating spermatids.

6. It is not clear how many independent testes (animals) were analyzed in each experiment (50 individual sperm/spermatids could be in a single seminal vesicle/testis in a single animal Fig.2.,3.,4.).

All experiments were based on testes from at least five independent animals. Corresponding information were included into the respective figure legends.

Acroblast formation typically begins in round spermatids; however, the localization pattern of Nep4-mNG does not coincide with that of GM130 in any stages of spermatogenesis.

We agree that the two signals do not overlap in early elongating spermatids. However, up to now we have not analyzed any additional stages of spermatogenesis regarding GM130 / Nep4-mNG co-distribution.

7. The differences in the localization pattern of Nep4-mNG and acrosome morphology between heterozygotes and homozygotes presented in Fig. 3 are not compelling.

We removed the corresponding statements for the CLEM analyses (Fig. 3A) and attenuated them for the fluorescence stainings (Fig. 3B). Quantification of the relevant morphological defects was done in Figs. 4 and 6.

8. If the antibody against Nep4 is effective in the testis, it raises the question of why it has not been utilized for colocalization studies with snky-GFP.

The Nep4 antibody requires a preincubation with SDS to produce a significant signal in tissue stainings; this procedure was not compatible with the GFP signal detection (see above). Corresponding information are provided in the Materials and Methods section.

9. In the analysis of sperm tips, it is necessary to measure and present the entire sperm nuclei, not just part of it (Fig. 5).

10. The evaluation of acrosome structural modifications illustrated in Figure 6 could be deceptive, given the considerable differences in nucleus lengths across the various genotypes presented in Fig. 6A. It is unclear whether the measurements correspond to mature sperm or elongated spermatids, indicating that the variations in acrosomal size might result from differing developmental stages. Furthermore, which genotype is utilized as the control in this experiment?

To ensure analysis of similar developmental stages, we analyzed sperm at exactly the same position within the respective testes (see Supplementary Fig. 8). Furthermore, we selected sperm with nuclei devoid of any organelles surrounding them, indicating a largely completed individualization. For the light microscopic analyses (Figs. 4, 5, S7) only individualized sperm that were not surrounded by cyst cells were selected.

w¹¹¹⁸ x UAS-nep4^{RNAi} was used as a control. Corresponding information were included into the figure legend.

11. Globozoospermia is a human disorder, and the conclusion that Nep4-mNG is akin to globozoospermia is an overreaching statement.

The statement has been removed from the abstract and attenuated in the discussion section.

Reviewer #3 (Remarks to the Author):

The manuscript from Buhr et al describes a study aimed at elucidating the role(s) of the peptidase Neprilysin 4 (Nep4) during spermatogenesis in *Drosophila*. Previous work from the authors and other groups had established that Nep4 is expressed during spermatogenesis and that Nep4 function is critical for male fertility. While Nep4-deficient sperm appear to develop normally and are capable of fertilizing eggs, they fail to promote any measure of embryonic development. However, detailed and in-depth analysis of Nep4 function during spermatogenesis has been lacking.

In order to proceed, the authors generated Nep4::mNG, an endogenously tagged variant of Nep4, in which the fluorescent marker mNeonGreen is attached to the C-terminus of the protein. This construct enabled them to both follow the localization of Nep4 during *Drosophila* spermatogenesis and to examine the mutant phenotype, as Nep4::mNG appears to be non-functional. The major findings made were as follows:

1. Nep4 exhibits a highly specific localization to the developing and mature sperm acrosome.
2. The acrosome is divided into several distinct domains, with Nep4::mNG localizing anteriorly to a second acrosomal marker, Snky-GFP.
3. Disruption of Nep4 function results in acrosomal abnormalities: the normally elongated acrosome takes on a more rounded morphology, and the partitioning of the acrosome and sperm tip into distinct domains are lost.
4. The severe phenotypes observed in Nep4::mNG homozygous males, which encodes a stable but non-functional peptidase, implies that the mutant phenotypes are indeed the result of absence of peptidase capacity.

This is a very well-executed study. The authors make highly competent use of different and complementary light and electron-microscopy techniques, along with careful quantitation of their data. The assignment of an acrosomal context for Nep4 function is novel and significant, and helps define clear avenues for future research. That said, the overall advance made in ascribing a defined function for Nep4 during spermatogenesis is rather minimal and does not go beyond focusing attention on the acrosome, without providing mechanistic insight into the manner by which Nep4 peptidase activity contributes to proper acrosome formation and organization, and why the observed acrosomal abnormalities cause such a strong disruption of the capacity of Nep4-deficient sperm to fertilize eggs and initiate embryogenesis. Sneaky mutant sperm are capable of penetrating eggs, indicating that the acrosome in *Drosophila* may not be essential for sperm penetration. However, the

sneaky mutant sperm nucleus fails to undergo decondensation after egg penetration. Further characterization of Nep4-deficient sperm following egg penetration would significantly enhance the manuscript.

To address this issue, we have conducted a series of new experiments. Analyses on the seminal receptacles revealed that Nep4::mNG sperm from homozygous males are present in the female storage organs, however, they are discarded rather quickly (relative to the heterozygous controls). Moreover, we found that the proportion of fertilized oocytes was significantly reduced under homozygous conditions, relative to the heterozygous control, and that none of the laid eggs initiated embryonic development, even if sperm entry had occurred.

These results are largely consistent with data obtained for a Nep4 mutant lacking most of the Nep4 extracellular domain (Ohsako et al., 2021) and indicate that the lack of Nep4 activity impairs sperm function at multiple levels and especially after transfer to females.

The described data are presented in a new Figure 7.

We thank all reviewers for their helpful and constructive comments! Based on these remarks, we did a series of additional experiments and were able to address most of the issues raised. Thus, the present manuscript represents a substantially amended version of the original paper, and we are confident that it now complies with the standards of "Communications Biology".

References

Ohsako, T. *et al.* The *Drosophila Neprilysin 4* gene is essential for sperm function following sperm transfer to females. *Genes Genet. Syst.* **96**, 21–00024 (2021).

Reviewers' comments:

Reviewer #1 (Remarks to the Author):

The authors have satisfactorily addressed all of my comments and concerns. I appreciate the effort they put in revising the manuscript and performing additional experiments to strengthen their conclusions. Therefore, I recommend the manuscript for publication in its current form.

Reviewer #2 (Remarks to the Author):

The originality of the current paper is constrained, as Ohsako, T. et al. have previously detailed the male sterile phenotype. The primary novelty of this paper lies in the localization of Nep4 to the acrosome; however, the absence of colocalization or co-staining with other acrosomal proteins or organelles and along with the inadequately presented new experimental data, undermines the robustness of the present research.

1. The effectiveness of RNA interference (RNAi) in the germline has not been clearly established. The efficiency of the RNAi line employed was previously assessed solely in somatic tissues, with no testing conducted in the testis as indicated in the referenced papers. KK RNAi lines may be effective in the germ line by using UAS-Dicer2 overexpression.

We assessed and confirmed a high testis-specific knockdown efficiency by western blot. Corresponding data are shown in a novel Supplementary Fig. 3.

The experiment demonstrating the effectiveness of RNA interference is questionable; it certainly does not show strong silencing efficiency. The objectivity of the evaluation is problematic. Ponceau S staining cannot be used for protein quantification in this form. In addition to presenting the entire Ponceau-stained membrane, not just a small part of it (which is barely visible), it is essential to detect a ubiquitously expressed protein with a specific antibody (loading control). The size of the protein allows the membrane to be cut, and the amount of protein applied could be verified with a well-established anti-actin or anti-tubulin antibody. Details of this experiment are missing from the materials and methods section. The molecular weight marker is barely visible.

As suggested, we reprobbed the membranes with antibodies detecting a ubiquitously expressed protein (GAPDH) and now include the corresponding staining as a loading control. The respective genotype-specific amounts of Nep4 protein are now normalized to this control. A significant knockdown was confirmed, relative to both the Gal4 ($w^{1118} \times bam$ -Gal4) and the UAS control ($w^{1118} \times UAS-nep4^{RNAi}$). RNAi mediated knockdown reduced Nep4 protein levels by about 70 %, which corresponds well with the reported efficiencies of other RNAi lines. In this regard, an analysis of the knockdown efficiency of a publicly available RNAi resource (TRiP collection, Harvard Medical School) revealed that >90% of the established lines exhibited residual gene expression of 25% or more (Perkins *et al.* 2015; Heigwer *et al.*, 2018).

Our new data are shown in an amended Supplementary Fig. 3. The Materials and Methods section was amended to include details on this experiment.

2. The reason for the observed fertility of w^{1118} -bam-Gal4 being twice that of w^{1118} alone remains

unclear.

In contrast to the *w¹¹¹⁸* and *Nep4::mNG* animals, which were reared at 22 °C (RT), all knockdown experiments (including the controls) were performed at 27 °C. This difference in temperature likely accounts for the increased number of adult offspring. Corresponding information are included in the Materials and Methods section.

It is not entirely clear what is meant by "including the controls."

The *w¹¹¹⁸* could be a control, similarly to the *Nep4-RNAi* line alone or the *bam-Gal4* line at 27 oC. What is the scientific evidence or explanation for the increased number of adult offspring at higher temperatures?

Why wasn't 27°C used for all experiments regardless of the genotype?

In this experiment, we used individual controls either for the homozygous *Nep4::mNG* (*w¹¹¹⁸* and heterozygous *Nep4::mNG* as specific controls) or for the *nep4* RNAi lines (*w¹¹¹⁸* x *bam-Gal4* and *w¹¹¹⁸* x *UAS-nep4^{RNAi}* as specific controls). The reason for this was based on the fact that the fertility and development of *Drosophila* is significantly influenced by temperature. Therefore, experiments were conducted at 27 °C only if absolutely necessary (i.e. UAS/Gal4 experiments). Since all experiments using the *Nep4::mNG* line were conducted at 22 °C, we also did the fertility assay at this temperature. By using appropriate controls for each group, we were able to perform intra-group comparisons, and the resulting effects were highly significant for both groups (*Nep4::mNG* knock-in and *nep4^{RNAi}* knockdown, Fig. 1). Information on the respective ambient temperatures are now included in the legend of Fig. 1.

Of note, we also counted the offspring of the RNAi-flies (*w¹¹¹⁸* x *bam-Gal4*, *w¹¹¹⁸* x *UAS-nep4^{RNAi}*, *bam* > *nep4^{RNAi}*, 27 °C) already at day 14 after mating, while the *Nep4::mNG* flies (22 °C) were still counted on day 18. As depicted below, under these conditions the difference between the respective controls for either the homozygous *Nep4::mNG* (*w¹¹¹⁸* and heterozygous *Nep4::mNG*) or for the *nep4* RNAi lines (*w¹¹¹⁸* x *bam-Gal4* and *w¹¹¹⁸* x *UAS-nep4^{RNAi}*) was absent, probably reflecting the accelerated *Drosophila* development with increased temperature (Al-Saffar et al., 1996; Dillon et al., 2007). While we have not included these data into the manuscript, they may still be helpful to explain the differences in question between the individual controls in Fig. 1.

There is a lack of data regarding the effectiveness of Nep4 downregulation (bam>nep4RNAi) through the use of anti-Nep4 antibodies or qRT-PCR. This is crucial for the characterization of the function of the nep4 gene.

We assessed and confirmed a high testis-specific knockdown efficiency by western blot. Corresponding data are shown in a novel Supplementary Fig. 3.

My answer to the first question applies here as well.

Please see reply to first question.

3. CRISPR technology enables the deletion of the gene segment that encodes the extracellular portion of the protein. However, it is uncertain whether tagging the protein results in a neomorphic effect or a gain-of-function mutation. A definitive deletion in the same gene region would provide stronger evidence.

The fact that the RNAi knockdown experiments phenocopied the effects of the tagging strongly indicates a loss-of-function effect. Corresponding information were included into the results section.

As established by Ohsako, T. et al., it has been demonstrated that Nep4 is crucial for normal fertilization; thus, the originality of the current paper must be substantiated through experimental evidence. A mutant with a deletion at the C-terminal could validate the essential role of Nep4 in maintaining normal fertility. Additionally, an N-terminally tagged variant of Nep4, created using CRISPR technology, would further corroborate these findings.

A corresponding mutant lacking large parts of the Nep4 extracellular domain is already available and has been analyzed by Ohsako *et al.*, in detail. The resulting effects were highly similar to the effects we observed for the homozygous Nep4::mNG fusion.

The materials and methods section fails to specify which gRNA was utilized; could this potentially lead to off-target effects in addition to tagging Nep4? The same gRNA might also be employed to produce a loss-of-function mutant.

Information on the utilized gRNA were included into the Materials and Methods section. Sequence analyses indicated no off-target activity of the selected gRNA. In addition, the fact that our *nep4* RNAi analyses replicated the main effects of homozygous Nep4::mNG largely ruled out off-target effects.

It is not addressed that male sterility is due to abnormal spermiogenesis or abnormal fertilization. Do the seminal vesicles or the spermatheca of the females used in the fertility test show Nep4-mNG signal in the homozygotes Nep4-mNG?

Seminal vesicles showing Nep4-mNG signals are depicted in Fig. 2B; presence of individualized sperm in the seminal vesicles indicates proper completion of spermatogenesis. The text was amended accordingly.

Further analyses on the seminal receptacles revealed that Nep4::mNG sperm from homozygous males are present in the female storage organs, however, they are discarded rather quickly (relative to the heterozygous control). Moreover, we found that the proportion of fertilized oocytes was significantly reduced under homozygous conditions, relative to the heterozygous control, and that none of the laid eggs initiated embryonic development, even if sperm entry had occurred. The resulting data are presented in a new Figure 7.

Answer accepted.

5. The colocalization of the antibody with Nep4-mNG has not been demonstrated in the same spermatids or sperm samples. What is the localization of the protein as determined by antibody staining during the earlier phases of spermatogenesis?

Since the Nep4 antibody requires a preincubation with SDS to produce a significant signal in tissue stainings, we were not able to do Nep4 / mNG co-stainings in the same samples (SDS incubation was incompatible with the mNG signal detection). Corresponding information are provided in the Materials and Methods section.

Given the high similarity of the Nep4 and mNG signal patterns (Figure S5), we concluded that Nep4::mNG exhibits a subcellular localization identical to that of the untagged protein. Thus, we rather analyzed the mNG signal in detail. No signal was detected in phases prior to early elongating spermatids.

Answer accepted.

6. It is not clear how many independent testes (animals) were analyzed in each experiment (50 individual sperm/spermatids could be in a single seminal vesicle/testis in a single animal Fig.2.,3.,4.).

All experiments were based on testes from at least five independent animals. Corresponding information were included into the respective figure legends.

The use of five independent animals in an experiment is very minimal, but a statement of the independent experiments in the materials and methods section would also be important for assessing reproducibility.

A corresponding statement was included into the Materials and Methods section.

Acroblast formation typically begins in round spermatids; however, the localization pattern of Nep4-mNG does not coincide with that of GM130 in any stages of spermatogenesis.

We agree that the two signals do not overlap in early elongating spermatids. However, up to now we have not analyzed any additional stages of spermatogenesis regarding GM130 / Nep4-mNG co-distribution.

It is interesting and important to find out the strong localization of Nep4 in the early elongating spermatids. Lysosomal, trans-Golgi, or basal body staining can help to specify better the nature of the signal.

We performed lysosomal (Arl8) and trans-Golgi (Golgin245) labeling in early elongating spermatids. While no co-localization was evident with the lysosomal marker, we observed at least partial co-localization between Nep4::mNG and Golgin245 at the trans-Golgi. Corresponding data are included in an amended Figure 2.

7. The differences in the localization pattern of Nep4-mNG and acrosome morphology between heterozygotes and homozygotes presented in Fig. 3 are not compelling.

We removed the corresponding statements for the CLEM analyses (Fig. 3A) and attenuated them for

the fluorescence stainings (Fig. 3B). Quantification of the relevant morphological defects was done in Figs. 4 and 6.

Answer accepted.

8. If the antibody against Nep4 is effective in the testis, it raises the question of why it has not been utilized for colocalization studies with snky-GFP.

The Nep4 antibody requires a preincubation with SDS to produce a significant signal in tissue stainings; this procedure was not compatible with the GFP signal detection (see above).

Corresponding information are provided in the Materials and Methods section.

Answer accepted.

9. In the analysis of sperm tips, it is necessary to measure and present the entire sperm nucleus, not just part of it (Fig. 5).

This question is not answered.

As is stated, “50 individual sperm cells isolated from the testes”. Using sperm from the seminal vesicles in this experiment would be more objective, including the entire sperm nucleus in the measurement, not just part of it.

We reanalyzed our current data set to measure the entire sperm nuclei in addition to the measurements of the anterior part that we had done previously. Corresponding size measurements are now presented in an amended Supplementary Figure 7. No significant differences were found in any of the genotypes analyzed.

10. The evaluation of acrosome structural modifications illustrated in Figure 6 could be deceptive, given the considerable differences in nucleus lengths across the various genotypes presented in Fig. 6A. It is unclear whether the measurements correspond to mature sperm or elongated spermatids, indicating that the variations in acrosomal size might result from differing developmental stages. Furthermore, which genotype is utilized as the control in this experiment?

To ensure analysis of similar developmental stages, we analyzed sperm at exactly the same position within the respective testes (see Supplementary Fig. 8). Furthermore, we selected sperm with nuclei devoid of any organelles surrounding them, indicating a largely completed individualization. For the light microscopic analyses (Figs. 4, 5, S7) only individualized sperm that were not surrounded by cyst cells were selected.

$w^{1118} \times UAS-nep4^{RNAi}$ was used as a control. Corresponding information were included into the figure legend.

Spermatogenesis is such a dynamic process; therefore, even in 24 hours, stages and distribution of cysts are changing, therefore, “exactly the same position” does not necessarily mean the same stages.

We agree that the way we have determined the developmental stages is not perfect. However, given the selection criteria applied (sperm were analyzed at exactly the same position within the respective testes (Supplementary Fig. 8); only sperm with nuclei devoid of any organelles surrounding them were used; only individualized sperm that were not surrounded by cyst cells were used) we assume that the analyzed sperm were mature. This assumption is supported by experimental data confirming identical lengths of the sperm nuclei of all genotypes analyzed in this study (Supplementary Fig. 7D,

D'), thus indicating similar stages of development (see also answer to question 9). The corresponding data are presented in an amended Supplementary Figure 7.

Moreover, considering the discrepancy between the actual measurements and the appearance of the nuclei in Fig. 6A, we reassessed our dataset and recognized that our initial approach of scaling the images based on the entire 3D data stack (bounding box) - rather than the individual 2D slices corresponding to each cell - created a perceptual bias when assessing lengths of nuclei and acrosomal tips.

We have now revised Fig. 6 to utilize scale bars calibrated specifically to each 2D section, accurately reflecting the dimensions observed within that plane. This adjustment eliminated the previously introduced distortion and ensured a more faithful and comparable visualization across genotypes.

The Materials and Methods section was amended to include this information.

11. Globozoospermia is a human disorder, and the conclusion that Nep4-mNG is akin to globozoospermia is an overreaching statement.

The statement has been removed from the abstract and attenuated in the discussion section.
Answer accepted.

Reviewer #3 (Remarks to the Author):

Following the revisions and additions made, I find that the manuscript has significantly improved and is now suitable for publication in your journal

References:

Al-Saffar, Z.Y. et al. (1996) Temperature and humidity affecting development, survival and weight loss of the pupal stage of *Drosophila melanogaster*, and the influence of alternating temperature on the larvae. *Journal of Thermal Biology* 21(5–6):389–396, doi:10.1016/S0306-4565(96)00025-3.

Dillon, M.E. et al. (2007) Life history consequences of temperature transients in *Drosophila melanogaster*. *J. Exp. Biol.* 210(16):2897–2904.

Heigwer, F. et al. (2018) RNA Interference (RNAi) Screening in *Drosophila*. *Genetics* 208(3):853–874.

Ohsako, T. et al. (2021) The *Drosophila* Neprilysin 4 gene is essential for sperm function following sperm transfer to females. *Genes Genet. Syst.* 96:177–186.

Perkins, L.A. et al. (2015) The Transgenic RNAi Project at Harvard Medical School: Resources and Validation. *Genetics* 201(3):843–52.